# On the Ability of Developers' Training Data Preservation of Learnware

**Hao-Yi Lei, Zhi-Hao Tan, Zhi-Hua Zhou**
National Key Laboratory for Novel Software Technology, Nanjing University, China
School of Artificial Intelligence, Nanjing University, China
`{leihy, tanzh, zhouzh}@lamda.nju.edu.cn`

## Abstract

The *learnware* paradigm aims to enable users to leverage numerous existing well-trained models instead of building machine learning models from scratch. In this paradigm, developers worldwide can submit their well-trained models spontaneously into a *learnware dock system*, and the system helps developers generate *specification* for each model to form a learnware. As the key component, a specification should characterize the capabilities of the model, enabling it to be adequately identified and reused, while preserving the developer's original data. Recently, the RKME (Reduced Kernel Mean Embedding) specification was proposed and most commonly utilized. This paper provides a theoretical analysis of RKME specification about its preservation ability for developer's training data. By modeling it as a geometric problem on manifolds and utilizing tools from geometric analysis, we prove that the RKME specification is able to disclose none of the developer's original data and possesses robust defense against common inference attacks, while preserving sufficient information for effective learnware identification.

## 1 Introduction

Various machine learning models have been applied into various aspects of modern life successfully [Butler et al., 2018, Jumper et al., 2021]. In conventional machine learning paradigm, developing a high-quality model for a new task from scratch requires a substantial amount of labeled data, expertise, and computational resources, while it is ideal if the solution of the new task can be built based on reusing existing models. Generally, source data is crucial for transferring and reusing existing models, however, concerns over data privacy and proprietary often hinder the sharing among developers.

The Learnware paradigm [Zhou, 2016, Zhou and Tan, 2024] offers a systematical way enabling users to build a new machine learning solution by exploiting existing well-trained models, rather than building a model from scratch. A learnware is a well-trained machine learning model facilitated with a *specification*, which characterizes the ability and specialty of the model to some degree, enabling the model to be adequately reused for new users without access to the original data used to train the model by its developer. Developers all over the world can submit their trained models into a *learnware dock system* spontaneously, and the system helps developers generate specifications, without access to the developer's training data. When a user wants to tackle her learning task, instead of starting from scratch, she can submit her requirement to the learnware dock system, which will then identify and return helpful learnware(s) to the user to reuse, such that the user can get a better performance than using her own data to train a model from scratch. Note that developers generally need to preserve their training data. To realize this attractive vision, the key challenge is: To solve new tasks, how to identify and even reassemble a few helpful models from the huge amount of learnwares accommodated in the dock system, without accessing each developer's training data?

38th Conference on Neural Information Processing Systems (NeurIPS 2024).

The specification is pivotal in the paradigm design [Zhou and Tan, 2024]. Recently, based on the RKME (reduced kernel mean embedding) specification [Zhou and Tan, 2024], many studies about learnware identification and reuse have been reported [Liu et al., 2024, Tan et al., 2023, Xie et al., 2023, Tan et al., 2024a, Wu et al., 2023]. Also, the *Beimingwu* learnware dock system has been developed and released [Tan et al., 2024b].

It is worth noting that, there lacks a theoretical analysis of the preservation ability of RKME specification for developer's training data. The theoretical analysis faces some significant challenges. Firstly, the RKME specification is generated by solving a non-convex optimization problem, whose possible solutions adhere to a nonlinear system of equations and, inherently, do not have a closed-form solution. This complexity makes direct analysis intractable. Moreover, the generation process of RKME is deterministic, but the size of RKME is small, where adding noise during this process often degrades its performance for learnware identification. Consequently, commonly used analytical tools for data privacy, such as Differential Privacy (DP), become inapplicable. To the best of our knowledge, there is little research analyzing whether and how synthetic data generated by deterministic algorithms can protect original data from being vulnerable to privacy attacks, especially considering that brute-force attacks often succeed against deterministic algorithms. In this paper, we prove that the RKME specification is able to protect the developer's original data from disclosure, and possess robust defense against common inference attacks, while preserving sufficient information for learnware identification. Our technique also provides a way to investigate the privacy preservation ability of synthetic data generated by deterministic algorithms. The main contributions of this work are summarized as follows:

- We prove that as the size of RKME specification decreases, the possibility that it discloses the original training data decreases at an exponential rate. Meanwhile, the ability of learnware identification is positively correlated with the size of RKME specification, and thus we also need a sufficient size. We prove that within a certain range of sizes, the RKME specification won't disclose any original training data for almost all datasets, while it remains sufficient information enabling effective learnware identification.

- We prove that RKME exhibits significant resistance to the two common types of data disclosure attacks: linkage and inference. We provide a method to measure data leakage of RKME and illustrate that the risk of RKME specification data leakage diminishes as the size of specification decreases. Within a certain range of sizes, the RKME can effectively resist these two types of attacks, while maintaining sufficient information for learnware identification. Moreover, we analyze that the above two ranges of sizes are heavily overlapped, implying that determining adequate sizes that enable learnware identification but preserving developer's data is practical.

## 2 Preliminary

In this paper, for the theoretical analysis of the preservation ability of RKME specification simplicity, we consider the following simplified learnware paradigm based on RKME specification. The learnware paradigm consists of two stages: submitting and deploying stages.

**Submitting stage.** In this stage, model developers submit their models to the learnware dock system. To better characterize these models, developers also provide the specification $R$ with each model to the system, which is generated by the RKME mechanism from the dataset $D$ used to train the model. A model together with its corresponding specification forms a learnware.

**Deploying stage.** In this stage, the user has a learning task and an unlabeled dataset $D_u$. To tackle the task, the user submits the task requirement $R_u$, which is generated by the RKME mechanism from the dataset $D_u$, and the system identifies helpful learnwares by comparing which learnware specifications are "close" to $R_u$. Subsequently, the user can solve her task by reusing these learnwares.

It is evident that the design of specifications is of great importance. To make the learnware identification (also called the search process) more precise, we hope the specification should contain some information about the data of the developers (or users). On the other hand, since developers face the challenge of uploading their specifications, it's crucial that the specification should protect the original data of the developer. The RKME generation process can be described as follows:

$$F(x_1, \cdots, x_n; \beta_1, \cdots, \beta_m; z_1, \cdots, z_m) = \left\| \frac{1}{n} \sum_{i=1}^{n} k(x_i, \cdot) - \sum_{j=1}^{m} \beta_j k(z_j, \cdot) \right\|_{\mathcal{H}}. \tag{1}$$

Here, $\{x_1, \cdots, x_n\}$ is the original dataset $D$ of the developer sampling from a certain distribution $\mathcal{P}$, with each data point $x_i = (x_i^1, \cdots, x_i^d)$ being $d$-dimensional. The RKME specification generated from $D$ is $R = \{\beta_1, \cdots, \beta_m; z_1, \cdots, z_m\}$ that minimizes Eq. (1) given the number of synthetic data $m$. We will focus on the set $Z = \{z_1, \cdots, z_m\}$, which consists of synthetic data sharing the same feature space with $x_i$. The $\beta_i \in \mathbb{R}$ are weights corresponding to each $z_i$. Here $k(\cdot, \cdot)$ is a kernel function, and $\| \cdot \|_{\mathcal{H}}$ is the norm in the reproducing kernel Hilbert space induced by $k(\cdot, \cdot)$. In this paper, we conduct our proofs using the Gaussian kernel $k(x, x') = \exp(-\gamma \|x - x'\|_2^2)$, where $\gamma > 0$, which is currently employed for generating RKME specifications. The generalizability of our proofs to other kernels will be discussed in the section 5.

The ability of RKME to search models can be characterized by its approximation of the Kernel Mean Embedding (KME) [Smola et al., 2007] of the original data distribution, as this reflects how well RKME characterizes the original data distribution. From Zhang et al. [2021], we arrive at the following conclusion:

**Lemma 2.1.** *Let the kernel $k$ satisfies that $k(\mathrm{x}, \mathrm{x}) \leq 1$ for all $\mathrm{x} \in X$ and any $\delta > 0$, we have*

$$\|\tilde{\mu} - \mu\|_{\mathcal{H}} \leq 2\sqrt{\frac{2}{m}} + 2\sqrt{\frac{1}{n}} + \sqrt{\frac{2 \log(1/\delta)}{n}}, \tag{2}$$

*with probability at least $1 - \delta$, where $\tilde{\mu}$ is RKME of $D$ and $\mu$ is KME of original data distribution $\mathcal{P}$.*

Existing experimental results have demonstrated that selecting $m = \sqrt{n}$ allows RKME to effectively search models, achieving success [Tan et al., 2023, Liu et al., 2024]. However, analysis of how RKME protects the data of both developers and users from a theoretical perspective is still lacking. Since the synthetic data $Z$ shares the same feature space with $X$, it is essential to investigate whether RKME contains any original data. Furthermore, whether the original data can be inferred from RKME through specific attack methods remains unknown. The following sections of this paper will explore these two perspectives to prove RKME's data preservation ability. All proofs are provided in the Appendix C.

## 3 Specification contains no original data

To analyze the RKME specification's ability to preserve original data, we first need to determine whether the specification contains original data. In this section, our objective is to quantify the consistency of synthetic data in RKME with the original data, as with most studies considering the privacy protection of synthetic data [Raghunathan, 2021, Abowd and Vilhuber, 2008]. Since RKME is generated through a deterministic algorithm, the above issue becomes more pressing than in randomized algorithms, where the output uncertainty can mitigate such concerns.

**Quantifying approach.** To quantify whether $\{z_1, \cdots, z_m\}$ in RKME contains any data from the original dataset $\{x_1, \cdots, x_n\}$, we analyze whether there exist $i \in \{1, \cdots, m\}$ and $j \in \{1, \cdots, n\}$ such that $z_i = x_j$, or more generally, $\|z_i - x_j\| \leq \delta$, which means that there are samples in RKME that are very close to some original data. We propose the following quantitative metric:

**Definition 3.1** (Consistency risk)**.** *We define the* consistency risk *of the RKME $Z$, generated from $n$ samples $D$ drawn from the distribution $\mathcal{P}$, containing original true data as:*

$$R_C(\mathcal{P}) \triangleq \mathbb{E}_{D \sim \mathcal{P}^n} \left( \mathbb{I}_{Z_\delta \bigcap D_\delta \neq \emptyset} \right),$$

*where $\mathbb{I}$ is the indicator function, and $Z_\delta \bigcap D_\delta \neq \emptyset$ indicates that, given $\delta$, there exists $i \in \{1, \cdots, m\}$ and $j \in \{1, \cdots, n\}$ such that $\|z_i - x_j\| \leq \delta$.*

As can be seen, the defined risk function $R$ represents the probability that the RKME generated from $n$ samples drawn from a potential distribution $\mathcal{P}$ may contain one of these $n$ original samples. The randomness here arises from the randomness of the sample set. It is evident that the value of $R_C(\mathcal{P})$ ranges between $[0, 1]$, with smaller values indicating a lower risk of RKME containing original samples. In the following, we will analyze the consistency risk for RKME specification.

**Technical overview.** To analyze the defined risk $R_C(\mathcal{P})$, the most natural approach would be to find a closed-form solution for $Z$ in relation to $D$, which would allow for a direct comparison of the elements of $Z$ and $D$. Unfortunately, solving the theoretical minimum of Eq. (1) is a nonlinear equation that does not have a closed-form solution. Therefore, this paper employs geometric analysis tools [Jost, 2008, Li, 2012], and analyzes the critical set of Eq. (1) to quantify the differences between the data in the synthetic set and the original sample set without solving the equation. To the best of our knowledge, we provide the first analysis based on geometric analysis tools to quantify privacy risk, which sheds light on data preservation analysis.

### 3.1 Consistenc risk evaluation

We start our analysis with the case of the data dimension $d = 1$, and as we will see, the case for any dimension $d$ can be reduced to the analysis of this scenario $d = 1$. To facilitate the use of geometric tools, we consider all data in the original dataset $D$ and its corresponding RKME $Z$ as wholes, namely as points in spaces $\mathbb{R}^n$ and $\mathbb{R}^m$, respectively. Let us denote them by $\mathbf{D} = (x_1, \cdots, x_n)$ in $\mathbb{R}^n$ and $\mathbf{Z} = (z_1, \cdots, z_m)$ in $\mathbb{R}^m$. The problem then translates into whether these two spaces have points with identical coordinate components. The upper bound of $R_C(\mathcal{P})$ is ensured based on the fact that in the space $\mathbb{R}^n$ where $\mathbf{D}$ resides, the set of $\mathbf{D}$ corresponding to $\mathbf{Z}$ in $\mathbb{R}^m$ with coordinate components equal to $\mathbf{D}$ is of small measure. In the remainder of the section, we will prove this.

We begin our analysis with the case of $\delta = 0$, focusing solely on whether any component of $\mathbf{Z}$ is strictly identical to $\mathbf{D}$. Our starting point is the correspondence between $\mathbf{D}$ and $\mathbf{Z}$. From this, we derive the following proposition:

**Proposition 3.2.** *The set of $\mathbf{D}$ in $\mathbb{R}^n$ that satisfies the condition of having multiple distinct $\mathbf{Z}$ which minimize Eq. (1) constitutes a set of measure zero.*

Proposition 3.2 allows us to consider the remaining $\mathbf{D}$ in $\mathbb{R}^n$ after removing a set of zero measures. These $\mathbf{D}$ correspond uniquely to a minimum value $\mathbf{Z}$. If we fix $\mathbf{Z}$, then, based on a similar idea as in the Implicit Function Theorem, we arrive at the following conclusion.

**Proposition 3.3.** *Given $\mathbf{Z}$ and $\{\beta_i\}_{i=1}^n$, consider the set $\mathcal{M}_Z$ defined as follows:*

$$\mathcal{M}_\mathbf{Z} = \left\{ (y_1, \cdots, y_n) \in \mathbb{R}^n \,\middle|\, F(y_1, \cdots, y_n; \mathbf{Z}) \leq F(x_1, \cdots, x_n; \mathbf{Z}), \forall (x_1, \cdots, x_n) \in \mathbb{R}^n \right\}.$$

*This set forms a manifold of dimension $n - 2m$. The subset $\mathcal{M}'_Z$, defined as*

$$\mathcal{M}'_\mathbf{Z} = \{(y_1, \cdots, y_n) \in \mathcal{M}_\mathbf{Z} \,|\, \exists i, j, z_i = y_j\},$$

*constitutes a submanifold of $\mathcal{M}_\mathbf{Z}$ with a dimension not exceeding $n - 2m - 1$.*

According to Proposition 3.2, we know that disregarding a set of zero measures, the $\mathcal{M}_\mathbf{Z}$ corresponding to different $\mathbf{Z}$ are disjoint. In each $\mathcal{M}_\mathbf{Z}$, the $\mathbf{D}$ with components identical to $\mathbf{Z}$ form a submanifold $\mathcal{M}'_\mathbf{Z}$ of dimension not higher than $n - 2m - 1$. Consequently, we obtain a representation for all possible points in $\mathbb{R}^n$ where $\mathbf{D}$ intersects with the generated $\mathbf{Z}$: they constitute a subset $\mathcal{M} = \bigcup_\mathbf{Z} \mathcal{M}'_\mathbf{Z}$.

Given that the dimension of $\mathbf{Z}$ is $m$, the Hausdorff dimension of $\mathcal{M}$(actually $\mathcal{M}$ is a manifold) is not greater than $n - 1$, which makes it a zero measurement set in $\mathbb{R}^n$. Thus, we conclude that for $\mathbf{D}$, which could possibly have the same coordinates as the generated $\mathbf{Z}$, is of measure zero in $\mathbb{R}^n$. From this observation, together with Proposition 3.2, we have the following theorem:

**Theorem 3.4.** *For any continuous original data distribution $\mathcal{P}$, when the size of the RKME set satisfies $m < \frac{n}{2}$, we have that the consistency risk $R_C(\mathcal{P}) = 0$.*

If $\mathcal{P}$ is a discrete distribution, then the above inference may not hold if the discrete values fall on $\mathcal{M}$. If $\mathcal{P}$ has very few possible values, $\mathbf{Z}$ will contain points from the original $\mathbf{D}$ when the number of points in $\mathbf{Z}$ is large (due to the presence of many identical samples in $\mathbf{D}$). In such cases, we find that by limiting $m$ to fewer than the possible values of $\mathcal{P}$, we can still prove $R_C(\mathcal{P}) = 0$ using some combinatorial techniques. Similar conclusions hold for mixed-type distributions.

In practical applications, it is desirable to ensure that RKME does not contain samples identical to those in the original dataset $D$. Therefore, we further explore whether RKME may include samples that are very close to those in the original data $D$, i.e., $\delta > 0$. A crucial aspect of addressing this issue involves the setting of $\delta$, which is significantly influenced by the inherent scale of the data and

the spacing between the data points. We will adopt a commonly used approach, selecting $\delta$ as the normalized value by the minimum spacing between different samples in the dataset.

Our idea of handling this scenario is fundamentally similar to that of $\delta = 0$. Given $\mathbf{Z}$, we similarly define $\mathcal{M}_{\mathbf{Z}}$, and $\mathcal{M}'_{\mathbf{Z}}$, is now defined as $\mathcal{M}'_{\mathbf{Z}} = \{(y_1, \cdots, y_n) \in \mathcal{M} \mid \exists i, j, |z_i - y_j| \leq \delta\}$. In this case, $\mathcal{M}'_{\mathbf{Z}}$ forms a measurable subset of $\mathcal{M}_{\mathbf{Z}}$ with a dimension of $n - 2m$. Based on the selection of $\delta$ as previously described, our objective is to estimate the ratio of the areas of $\mathcal{M}'_{\mathbf{Z}}$ and $\mathcal{M}_{\mathbf{Z}}$. To achieve this, we estimate the local curvature, perform local linearization, and use ideas similar to isoperimetric inequalities for the estimation. We arrive at the following conclusion.

**Theorem 3.5** (Bound of consistency risk). *For any continuous original data distribution $\mathcal{P}$, for RKME with $m$ synthetic data, we have*

$$R_C(\mathcal{P}) < \mathcal{O}\left(\left(\frac{1}{e}\right)^{n-2m}\right). \tag{3}$$

**Remark** We believe that the privacy protection afforded by RKME results from the many-to-one correspondence between the original sample set and the generated RKME. This is essentially a loss of individual member information caused by compression. However, not all synthetic data generation processes like this can achieve similar ideal effects. We have proved that if we choose the Laplacian kernel ($k(x, y) = \exp(-\gamma \|x - y\|_1)$) instead of the Gaussian kernel in Appendix C.8, Theorem 3.4 would no longer hold. In fact, with the Laplacian kernel, we can prove that the synthetic data induced by the corresponding RKME will definitely contain data identical to the original sample set. This underscores the rationality of choosing the Gaussian kernel RKME as our specification.

For cases where the data dimension $d > 1$, we can similarly define $\mathcal{D} = \{(\|x_1\|, \cdots, \|x_n\|) \in \mathbb{R}^n\}$ and $\mathcal{Z} = \{(\|z_1\|, \cdots, \|z_m\|) \in \mathbb{R}^m\}$. Using the same approach, and based on Thm. 3.5 and the inequality $\|x - y\| \leq |\|x\| - \|y\||$, we can derive conclusions for the $d$-dimensional case. The only difference is a change in order, as given by the formula: $R_C(\mathcal{P}) < \mathcal{O}\left(\left(\frac{1}{e}\right)^{dn-2dm-m}\right)$.

## 3.2 Data preservation and search ability

As indicated in Thm. 3.5, we observe that the consistency risk decreases as the size of the RKME specification decreases, which means that selecting a smaller number of synthetic data points $m$ can better ensure that RKME does not contain samples closely resembling the original data. If we represent the data protection capability of RKME in terms of not containing original data using $1 - R_C(\mathcal{P})$, and the ability of RKME for search derived in Lemma 2.1, we can illustrate the resulting trade-off, as shown in the following graph.

In this Pareto frontier, it is challenging to define the exact point of Pareto optimality. This difficulty arises from the differing scales of measuring data protection capabilities and the gap in RKME's approximation of the actual data distribution KME, which represents search efficiency. It is hard to set a rule to find the Pareto optimum due to these distinct measurement scales. However, fortunately, in existing works, we have observed that when the size of the RKME specification $m$ is larger than $\sqrt{n}$, the specification achieves satisfactory results in the Learnware's search tasks. In this paper, we propose the following corollary:

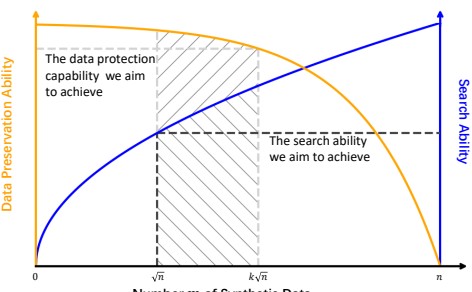

Figure 1: Trade-off between data consistency preservation and search ability.

**Corollary 3.6.** *If we choose $m \leq k\sqrt{n}$, where $k \geq d$, for our defined $R_C(\mathcal{P})$, we obtain the following equation:*

$$R_C(\mathcal{P}) < 0.001. \tag{4}$$

*This implies that we are 99.9% confident that for any dataset $D$ sampled from the distribution $\mathcal{P}$, the generated RKME will contain no synthetic data points that are close to the original samples in $D$.*

The conclusion above offers a flexible approach to selecting the number of synthetic data points $m$ in RKME. As illustrated in Fig. 3.2, the shaded area encompasses the range that allows RKME to

achieve efficient search capabilities and robust data protection, specifically in terms of not containing data closely similar to the original dataset.

# 4 Specification defends data inference

Whether a learnware's specification contains original data is not the only concern for its data protection ability. Stadler et al. [2022] note that synthetic data may not withstand traditional data attacks such as *linkage* [Elliot et al., 2018, Sweeney, 2002], and *attribute disclosure* [Elliot et al., 2018, Machanavajjhala et al., 2007]. Common defenses against these attacks usually involve formal privacy guarantees during the training process of the generative model to prevent breaches [Abowd and Vilhuber, 2008, Bindschaedler et al., 2017], or the addition of noise to the synthetic data generation process to satisfy differential privacy criteria [Xin et al., 2022, Jordon et al., 2018]. However, there is still a lack of research on whether synthetic data generated by a deterministic mechanism can naturally resist these attacks without extra noise.

RKME, in addition to potentially containing explicit original data, may also implicitly contain certain information about the data, which could be inferred under these two types of attacks. We aim to prove that RKME can protect against such inferences from the original data.

**Quantifying approach.** We suggest that the RKME mechanism, as a deterministic generation mechanism, serves as a *data anonymization solution* [Kuppa et al., 2021]. To verify whether the generation of the specification acts as an effective anonymization mechanism, our objective is to quantify whether such a specification can address the data leakage risks that data anonymization techniques are designed to mitigate. We quantified the data leakage risks associated with two types of concerns: *linkability* and *inference ability*, which correspond to the defenses against the linkage attack and the disclosure of attributes, respectively. For the risks of linkability and inference, we model each privacy concern as an adversary tasked with determining whether, given the specification or the original dataset, information about a component $x_t$ of a sample $x$ in $X$, or its attributes, can be inferred. For each adversary, we define a data leakage risk to measure the risk of the adversary inferring the sample $x$ or its attributes from $X$ after the RKME $Z$ of dataset $D$ has been released.

## 4.1 Linkability risk evaluation

In considering data sharing with privacy protection characteristics, a primary concern is the risk of linkability, which corresponds to an adversary conducting a linkage attack on the data. A linkage attack aims to link a target record to a single record or a group of records in a sensitive dataset. Specifically, we model the linkage attack as an adversary attempting to demonstrate whether a record is present in a sensitive dataset.

**Quantifying approach.** We suggest that the RKME generation mechanism, as a deterministic generation mechanism, serves as a *data anonymization solution* [Kuppa et al., 2021]. To verify whether the generation of the specification acts as an effective anonymization mechanism, we aim to quantify whether such a specification can address the data leakage risks that data anonymization techniques were designed to mitigate. We quantified the data leakage risks associated with the two types of leakage concerns: linkability and inference. These two are universally important de-anonymization techniques. Additionally, the synthetic data generated by RKME and the original data exist within the same feature space, making linkage attacks a potentially successful form of attack within the learnware paradigm, and thus a focal point of our analysis. Moreover, an attacker attempting to compromise the original data through RKME can only access a specific RKME, rendering many attacks ineffective in the learnware paradigm (such as requiring multiple queries).

**Linkability privacy game.** Following the works of [Pyrgelis et al., 2017, Yeom et al., 2018, Stadler et al., 2022], we model the risk of linkability as a membership privacy game between an adversary $\mathcal{A}$ and a challenger $\mathcal{C}$. In the learnware paradigm, learnware developers, who are also holders of the original data $D$, are considered as the challengers in the membership privacy game, where $Z$ generated from $D$ is visible to the adversary $\mathcal{A}$. The objective of the adversary $\mathcal{A}$ is to infer whether a target record $x_t$, chosen by the adversary, is present in the original dataset $D$, based solely on the knowledge of $Z$ and some prior knowledge $\mathcal{P}$.

The game 4.1 models a privacy game to assess the potential linkage attack on RKME synthetic data. Initially, $\mathcal{A}$ selects a target record $x_t$ and sends it to $\mathcal{C}$. Then $\mathcal{C}$ independently and identically draws a dataset $D$ of size $n-1$ from the data distribution $\mathcal{P}$, and chooses a secret bit $s_t \sim \{0,1\}$. If $s_t = 0$, $\mathcal{C}$ samples another record $x^*$ from the distribution, excluding the target record, and adds it to $D$ (simulating the scenario where the original dataset does not contain the target). If $s_t = 1$, $\mathcal{C}$ adds the target record $x_t$ to $D$ (simulating the scenario where the original dataset contains the target). Subsequently, $\mathcal{C}$ generates the corresponding RKME $Z$ using the RKME generation mechanism from $D$, and randomly sends either the original dataset $D$ or its corresponding RKME $Z$ to the adversary. Upon receiving the dataset, the adversary $\mathcal{A}$ guesses whether the target $x_t$ is in the original dataset $D$ through $\hat{s}_t \leftarrow \mathcal{A}^{\mathcal{L}}(D, b, x^t, \mathcal{P})$. If $\hat{s}_t = s_t$, it is considered that the adversary has successfully carried out a linkage attack.

---

**Algorithm 1** Linkability Privacy Game

---

1: $\mathcal{A}$ selects a target record $x_t$ from the data space $X$
2: $\mathcal{A}$ sends $x_t$ (or $\tilde{x}^t$) to $\mathcal{C}$
3: $\mathcal{C}$ samples a dataset $D$ of size $n-1$ from $\mathcal{P}$
4: $\mathcal{C}$ randomly chooses a secret bit $s_t$ from $\{0,1\}$
5: **if** $s_t = 0$ **then**
6:     $\mathcal{C}$ samples a random record $x^*$ from $\mathcal{P}_{\backslash x_t}$
7:     $\mathcal{C}$ adds $x^*$ to $D$
8: **else**
9:     $\mathcal{C}$ adds the target record $x_t$ to $D$
10: **end if**
11: $\mathcal{C}$ generates RKME $Z$ with $D$
12: $\mathcal{C}$ randomly chooses a public bit $b$ from $\{0,1\}$
13: **if** $b = 0$ **then**
14:     $\mathcal{C}$ sends $D$ to $\mathcal{A}$
15: **else**
16:     $\mathcal{C}$ sends $Z$ to $\mathcal{A}$
17: **end if**
18: $\mathcal{A}$ receives either $D$ or $Z$ and the public bit $b$
19: $\mathcal{A}$ uses its linkability algorithm $\mathcal{A}^{\mathcal{L}}$ to guess $\hat{s}_t$
20: $\mathcal{A}$ makes a guess $\hat{s}_t = \mathcal{A}^{\mathcal{L}}(X, b, x_t, \mathcal{P})$, $X = D$ or $Z$
21: **if** $\hat{s}_t = s_t$ **then**
22:     Adversary $\mathcal{A}$ wins in the linkage attack
23: **else**
24:     Adversary $\mathcal{A}$ fails to carry out a linkage attack
25: **end if**

---

**Definition 4.1** (Linkage risk). *We define the* linkage risk *of dataset $X$ during the linkage privacy game as*

$$R_L(X) \triangleq \sup_{x_t \in \mathbb{R}^n} \left( 2\mathrm{P}\left[\mathcal{A}^{\mathcal{L}}\left(X, b, x^t, \mathcal{P}\right) = s_t\right] - 1 \right)$$

*where $X = D$ or $Z$, and the probability space is composed of the randomness of the target, the randomness of the secret bit, and the adversary's method of guessing.*

Analyzing the linkage risk as defined above hinges on a critical examination of $\mathcal{A}^{\mathcal{L}}(X, b, x^t, \mathcal{P})$. This expression reflects the potential strategies that an adversary $\mathcal{A}$ might employ to guess whether a target record exists in the original dataset $D$. The risk of linkage attacks on RKME can vary depending on the strategy employed. Since the RKME generation mechanism is a deterministic algorithm, the most formidable attack strategy an adversary could deploy can be *brute-force attack*.

**Adversarial strategy.** When the adversary receives the information $X$ and $b$ provided by the challenger, the approach varies according to the value of $b$. If $b = 0$, the adversary has the original dataset $D$ and merely needs to check if the target $x_t$ is in $D$. When $b = 1$, the adversary receives the RKME $Z$ generated from $D$ and knows the data's prior distribution $\mathcal{P}$. The adversary then employs a brute-force attack to construct all possible datasets $D'$, collectively denoted as $M$, each of which corresponds to RKME $Z$. The probability of these $D'$ being the actual dataset under $\mathcal{P}$ varies, and the adversary can calculate the measure $dP(D|\mathcal{P})$ for each $D'$ in $M$. They can assign a weight $\frac{dP(D|\mathcal{P})}{\int_M dP(D|\mathcal{P})}$ to each $D'$ based on this measure, and then randomly select one $D'$ as the inferred true dataset using this weight. The adversary checks if the target $x_t$ is in this $D'$ and makes a guess.

**Risk evaluation.** Similarly to the method used for analyzing the consistency risk in the previous section, for a sample set $D = \{x_1, \cdots, x_n\}$ and its corresponding RKME $Z = \{z_1, \cdots, z_m\}$, we map them to points in $\mathbb{R}^n$ and $\mathbb{R}^m$ respectively, as $\mathbf{D} = \{(\|x_1\|, \cdots, \|x_n\|)\}$ and $\mathbf{Z} = \{(\|z_1\|, \cdots, \|z_n\|)\}$. Similarly, we find that the $\mathcal{M}$ formed by $\mathbf{Z}$ constitutes a manifold.

For the linkage risk term $2\mathrm{P}\left[\mathcal{A}^{\mathcal{L}}(X, b, x^t, \mathcal{P}) = s_t\right] - 1$, it can be interpreted as the difference between the adversary $\mathcal{A}$'s true positive rate and false positive rate, expressed as

$$\mathrm{P}\left[\hat{s}_t = 1 \mid s_t = 1\right] - \mathrm{P}\left[\hat{s}_t = 1 \mid s_t = 0\right]$$

The first term $P[\hat{s}_t = 1 \mid s_t = 1]$ corresponds to the probability of randomly selecting a $\mathbf{D}' \in \mathcal{M}$ weighted on the manifold $\mathcal{M}$, where $\mathbf{D}'$ contains coordinate components identical to $\mathbf{Z}$. Expanding this term, we have the following.

$$P[\hat{s}_t = 1 \mid s_t = 1] = \int_{\mathcal{M}} \frac{dP(D|\mathcal{P})}{\int_{\mathcal{M}} dP(D|\mathcal{P})} \left( \mathbb{I}_{Z \cap D \neq \emptyset} \right)$$

Similarly to the proof of Thm. 3.5, we can bound the first term. For the second term $P[\hat{s}_t = 1 \mid s_t = 0]$, we need to estimate the points in $\mathcal{P}$ that might generate $\mathbf{Z}$ but do not contain the target record $x_t$. We can provide an upper bound using the isoperimetric inequality. The deductions made above can be summarized in the following theorem.

**Theorem 4.2** (Bound of linkage risk). *When the adversary employs a brute-force attack, the linkage risk is bounded as follows:*

$$R_L(Z) < O\left( \frac{(2dm)!}{(dn - 2dm)!} \right). \tag{5}$$

**Remark** Our assessment of the risk of linkability is based on a worst-case evaluation, which is applicable to any target. This approach differs from many previous studies that have focused more on the average-case scenario. However, previous studies have shown that privacy risks associated with data sharing are not uniformly distributed throughout the population [Kulynych et al., 2022, Long et al., 2020, Rocher et al., 2019]. Our analysis of the worst-case scenario aligns more closely with practical needs, as we aim to ensure sufficient privacy protection for each individual data point.

## 4.2 Inference risk evaluation

The risk of linkability is not the only concern about data leakage in data sharing processes. Data anonymization mechanisms also protect individuals in the original data from attribute disclosure, which corresponds to an inference attack. The risk of inference describes the concern that an adversary might deduce the value of an attribute from the other attributes [El Emam and Alvarez, 2015].

As illustrated in the Game 4.2, this approach differs from the previous linkability privacy game in that the adversary only has access to a subset of the attributes of the target record, $x^1, \cdots, x^{s-1}$, and attempts to infer an unknown sensitive attribute value $x^s$. At the start of the game, the adversary randomly selects a target record from $\tilde{X}$, which is a collection of samples from $X$ with their sensitive attributes removed. Upon receiving partial information of this target record, the challenger assigns it a secret value $x^s \leftarrow \phi(\tilde{x}^t)$, where $\phi$ denotes the projection of a partial record from $\tilde{X}$ into the domain of the sensitive attribute based on its distribution. The challenger then merges the partial attributes provided by the adversary with the secret value assigned to form a complete sample in $X$, following which the same privacy game as in the linkability case is played. The adversary's final information comprises the dataset $X$ and a public bit $b$. Using this information, the adversary makes a guess about the target's sensitive attribute value $\hat{x}^s$. If the guess falls within our acceptable tolerance range, the adversary is considered to have won.

---

**Algorithm 2** Inference Privacy Game

1: $\mathcal{A}$ selects a partial target record $\tilde{x}^t$ from $\tilde{X}$, where $\tilde{X}$ contains samples from $X$ with some attributes removed.
2: $\mathcal{A}$ sends $\tilde{x}^t$ to $\mathcal{C}$; $\mathcal{C}$ assigns a sensitive value $x^s$ to $\tilde{x}^t$ using $\phi$, forming $x_t = (\tilde{x}^t, x^s)$
3: $\mathcal{C}$ samples a dataset $D$ of size $n - 1$ from $\mathcal{P}$
4: $\mathcal{C}$ randomly chooses a secret bit $s_t$ from $\{0, 1\}$
5: **if** $s_t = 0$ **then**
6:      $\mathcal{C}$ samples a random record $x^*$ from $\mathcal{P} \backslash x_t$
7:      $\mathcal{C}$ adds $x^*$ to $D$
8: **else**
9:      $\mathcal{C}$ adds the complete target record $x_t$ to $D$
10: **end if**
11: $\mathcal{C}$ generates RKME $Z$ from $D$
12: $\mathcal{C}$ randomly chooses a public bit $b$ from $\{0, 1\}$
13: **if** $b = 0$ **then**
14:      $\mathcal{C}$ sends $D$ to $\mathcal{A}$
15: **else**
16:      $\mathcal{C}$ sends $Z$ to $\mathcal{A}$
17: **end if**
18: $\mathcal{A}$ receives either $D$ or $Z$ and the public bit $b$
19: $\mathcal{A}$ uses its inference algorithm $\mathcal{A}^{\mathcal{I}}$ to guess $\hat{s}_t$
20: $\mathcal{A}$ makes a guess $\hat{s}_t = \mathcal{A}^{\mathcal{I}}(X, b, x_t, \mathcal{P}), X = D$ or $Z$
21: **if** $\hat{s}_t = s_t$ **then**
22:      Adversary $\mathcal{A}$ wins in the inference attack
23: **else**
24:      Adversary $\mathcal{A}$ fails to carry out an inference attack
25: **end if**

---

**Definition 4.3.** *We define the* inference risk *in the Inference Privacy Game as*

$$R_I(X) \triangleq \sup_{x^s \in \mathbb{R}} P\left[\hat{x}^s = x^s \,|\, s_t = 1\right] - P\left[\hat{x}^s = x^s \,|\, s_t = 0\right]$$

**Adversarial strategy.** To estimate $R_I$, we need to consider the strategy of guessing of the adversary. The adversary makes an estimate of the sensitive attribute value $x^s$ in $x_t$ based on the RKME $Z$ released by the Challenger, the public bit $b$, and the known partial attribute values $\tilde{x}^t$. Let us first consider the case where $b = 0$, where the Challenger releases the original dataset. In this scenario, the adversary can deduce the missing attribute value through record linkage [Drechsler and Reiter, 2010, Machanavajjhala et al., 2007, Reiter and Mitra, 2009]. If the adversary can link the partial information of the target with a unique sample in the original dataset (that is, there is only one sample whose partial information matches that of the target), then the missing value can be successfully reconstructed. In this case, $P[\mathcal{A}^{\mathcal{I}}(X, b, x^t, \mathcal{P}) = x^s \,|\, s_t = 1] = 1$.

When direct inference using linkage fails, the challenger must seek alternative methods to conjecture the target record. Similarly, due to the deterministic mechanism of RKME, we still analyze the brute-force attack. Specifically, like the previous linkage, the adversary first finds all possible sample sets that correspond to RKME $Z$ using a brute force attack. Among these sets, there may be some where a subset of attributes of certain samples matches the target record. The adversary then selects the original sample set from these subsets with partially matching information, using prior probabilities similar to the brute-force attack for linkage. Through an analysis similar to our previous approach, we have the following theorem.

**Theorem 4.4** (Bound of inference risk). *When the adversary employs a brute-force attack, the inference risk is bounded as follows*

$$R_I(Z) < O\left(\frac{(2m)!}{e^{(n-2m-1)}(n-1)!}\right) \tag{6}$$

### 4.3 Data preservation and search ability

Analogously to the analysis of the consistency risk, we hope that RKME can withstand linkage and inference attacks while still providing effective search capabilities. Regarding search ability, we continue to use the characterization of how the RKME generated from dataset $D$ approximates its original distribution $\mathcal{P}$ with varying numbers of synthetic data, as described in Lemma 2.1. We represent the protective capacity of RKME for dataset $D$ against the two types of attacks using $R_L(D) - R_L(Z)$ and $R_I(D) - R_I(Z)$, respectively. These represent the reduction in linkage and inference risks when publishing RKME instead of the original data. Based on Thm. 4.2 and Thm. 4.4, we propose the following corollary:

**Corollary 4.5.** *If we choose $m \leq \frac{n}{2}$, for $R_L(D) - R_L(Z)$ and $R_I(D) - R_I(Z)$, we obtain the following equation:*

$$R_L(D) - R_L(Z) \geq 0.999 \tag{7}$$
$$R_I(D) - R_I(Z) \geq 0.999 \tag{8}$$

*This implies that we have 99.9% confidence that RKME can defend the linkage and inference attack.*

To ensure that RKME maintains a risk level below our tolerance threshold (i.e., 0.001%) for consistency risk, as well as for linkage and inference risks, and still achieves satisfactory search efficiency, it suffices to select the number of points within the range $(\sqrt{n}, \min(k\sqrt{n}, \frac{n}{2}))$. This range offers flexibility in adjusting the number of points $m$. When greater precision in search is required, we can opt for a larger value of $m$ within this interval. In contrast, when a higher degree of data protection is desired, a smaller value of $m$ can be chosen within the same range.

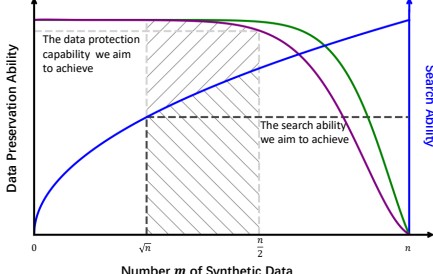

Figure 2: Trade-off between the ability of data linkage (inference) protection and search ability.

# 5 Discussion

In our work, we have provided proofs only for the Gaussian kernel; however, our method can be extended to analyze a broad class of kernels and yield similar conclusions regarding their data protection capabilities. For kernels that exhibit non-rationality and analyticity (such as the Sigmoid kernel $K(x,y) = \tanh(\gamma x^T y + r)$ and the Cauchy kernel $K(x,y) = \frac{1}{1+\gamma|x-y|^2}$), they can be treated similarly to the Gaussian kernel by considering samples as bundles of synthetic data within the sample space. The difference lies in that, due to the specific forms of these kernels, the risk estimates we calculate during our analysis will vary, and the derivative estimates in the proof process may require re-evaluation. Although each kernel will still require its own specific analysis, the approach provided in this paper is generally applicable. Extending this framework to prove more robust results for a broader class of kernels will be part of our future work.

The optimization problems involving Gaussian kernels are non-convex and non-rational, making theoretical analysis intractable under traditional tools. Analyses of optimization problems using the Gaussian kernel often rely on numerical experiments for validation. For the first time, we have made it possible to analyze the optimal solution using geometric analysis techniques to analyze the data protection capability of the RKME specification. This approach not only applies to the privacy analysis of RKME as a specific form of synthetic data, but it also provides insights for analyzing similar nonlinear non-convex optimization problems involving Gaussian kernels. Applying this technique to broader contexts will be part of our future work. The main limitation of this analytical method is that the upper bound may not be optimal due to multiple steps of restrictions and relaxations. However, pursuing a tighter upper bound remains theoretically significant.

In terms of relevant privacy theories, differential privacy [Dwork, 2006] (DP) is the most widely used technique, with its privacy protection characteristics derived primarily from the randomness introduced by additional noise. However, for the RKME specification in our study, due to its size, the well-known privacy-utility trade-off in differential privacy [Alvim et al., 2012] particularly pronounced after we add noise to the RKME mechanism, which can significantly degrade performance in learnware identification. This means that applying existing DP techniques to analyze privacy in RKME is quite challenging. On the other hand, due to the extensive data compression inherent in the RKME generation process, we believe that it possesses the data protection capabilities necessary within the learnware paradigm. Therefore, this paper also offers a perspective on how a compressive deterministic algorithm can achieve privacy protection without relying on DP methods.

Another important perspective for future work is to establish sufficient criteria for when specifications in learnware can provide strong protection for the developer's original data. This paper evaluates the risks associated with RKME containing original data, consistency risk, and the exposure risks under two common types of attacks: linkage risk and inference risk. We prove the necessity of protecting original data through the RKME specification induced by the Gaussian kernel. Due to the deterministic nature of the RKME generation mechanism, some attacks relying on randomness, such as multiple queries, are ineffective against RKME, and the evaluation criteria presented in this paper are broadly applicable. However, seeking more general evaluation standards and investigating what types of specifications can effectively protect the developer's original data under these criteria will be a key focus of our future work.

# 6 Concluding remarks

This paper presents a theoretical study about the ability of developer's data preservation of the RKME specification, which was recently proposed for building learnware specification [Zhou and Tan, 2024] and used in the Beimingwu learware dock system [Tan et al., 2024b]. By leveraging geometric analysis techniques, we prove that as the size of RKME specification decreases, the ability of the developer to preserve data increases, that is, the possibility of exposing the developer's original data decreases, and the ability to defend against the two commonly encountered attacks, i.e., linkage attack and inference attack, increases. Moreover, there exists a broad range of specification sizes that endow the above properties and enable effective learnware identification. Note that this work also offers a new perspective on the data preservation ability of reduced sets and the corresponding analysis of deterministic algorithms.

## Acknowledgments

This research was supported by NSFC (62250069). The authors would like to thank Shuo Zhang, Rui-Ming Liang, Yang Zhang for very helpful discussions. We are also grateful for the anonymous reviewers for their valuable comments.

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

# A Related work

## A.1 Related work of Learnware

The learnware paradigm [Zhou, 2016, Zhou and Tan, 2024] offers a systematic approach to managing well-trained models and leveraging their capabilities to assist users in solving their tasks, rather than training a model from scratch. A learnware consists of a well-trained model accompanied by a specification that describes its capabilities, with this specification being the central component of the learnware. Based on the RKME specification, which uses a reduced set to sketch the distribution of the task data, Wu et al. [2023] proposed to identify helpful learnwares by matching the task distribution. Zhang et al. [2021] extended it to handle user tasks with unseen parts. To efficiently recommend learnwares among numerous learnwares, Liu et al. [2024] suggested evolving the specification with other learnwares for more accurate recommendations and construct the tree structure for managing learnwares for efficient learnware identification, and Xie et al. [2023] proposed using minor representative learnwares as anchor learnwares to speed up learnware identification without traversing the whole system. For the models and user tasks share the different feature space, Tan et al. [2024a] first considers the heterogeneous feature space scenario, but it assumes that the original training data is accessible, and auxiliary data across the entire feature space is collected. To relax this strong assumption of data accessibility, Tan et al. [2023] investigates the organization and utilization of a heterogeneous learnware dock system without requiring access to the original data or auxiliary data across the feature space.

Based on the above research, the first learnware docking system, Beimingwu [Tan et al., 2024b], was recently released. This system streamlines the entire learnware process and offers a highly scalable architecture, facilitating future algorithm implementation and experimental research. Given these progresses, the RKME specification plays a crucial role within the learnware paradigm. However, despite its effectiveness, a theoretical analysis of the preservation capability of the RKME specification for the developer's training data is still lacking. Proving that the specification can protect the developer's training data not only safeguards both developers and users of learnware but also ensures the rationality of learnware specifications. This paper proves that the RKME specification can scarcely contain any of the developer's original data and provides robust defense against common inference attacks, while preserving sufficient distribution information for effective learnware identification.

## A.2 Related work of privacy

As the interaction of data becomes increasingly prevalent in various machine learning scenarios, the issue of sharing data while safeguarding data privacy has emerged as a critical concern. An important solution to this challenge is the release of synthetic data, generated through specific mechanisms. This approach manages to achieve an effectiveness comparable to original data in certain aspects (such as model training, data analysis, etc.), while simultaneously protecting the privacy of original data [Drechsler and Reiter, 2010, Bellovin et al., 2019, Arnold and Neunhoeffer, 2020, Stadler et al., 2020, England, 2022, Stadler et al., 2022, Xu et al., 2019].

Previously, it was commonly assumed that synthetic data, being 'artificial', have no direct link to real data. Therefore, in earlier work, it was believed that synthetic data generated by generative models, such as GANs [Goodfellow et al., 2014, Frid-Adar et al., 2018, Arjovsky et al., 2017, Salimans et al., 2016] and VAE [Kingma and Welling, 2014, Yan et al., 2016], have the capability to protect the privacy of original data. Their effectiveness was typically demonstrated through experimental similarity tests between real and synthetic records to assess the privacy risks associated with synthetic datasets [Choi et al., 2017, Yale et al., 2019a,b].

Unfortunately, Chen et al. [2020] pointed out that the aforementioned generative models could expose original data under certain privacy attacks. Stadler et al. [2022] further noted that not all synthetic data can withstand traditional attacks on data, such as linkage [Elliot et al., 2018, Sweeney, 2002], and attribute disclosure [Elliot et al., 2018, Machanavajjhala et al., 2007]. To counter this issue, existing efforts [Xin et al., 2022, Jordon et al., 2018, Harder et al., 2021] have applied differential privacy (DP) [Dwork et al., 2006] to develop differentially private data generators, as DP is the de facto privacy standard which provides theoretical guarantees of privacy leakage. Data produced by DP-generators can then be applied to various tasks, such as data analysis, visualization, training privacy-preserving classifiers, etc.

However, although DP appears to be a theoretically guaranteed privacy solution, Alvim et al. [2018], Zhao et al. [2020] have pointed out that DP involves a tradeoff between privacy and utility, and in some cases, this tradeoff can lead to a significant decrease in utility. Particularly, in the context of learnware specification, due to the requirement for usability, we demand that the number of synthetic data be much smaller than the number of original data, which results in a severe decrease in utility when applying DP, making it unsuitable for our framework. We attempt to demonstrate that significant data compression can also lead to data privacy, similar to the idea of dataset condensation (DC) [Loo et al., 2023, Wang et al., 2018]. However, in the work on DC, the privacy protection properties of synthetic data are primarily assessed through experimental validation. In our work, we aim to analyze this aspect theoretically, brought about by reduced synthetic data by modeling it as a low-dimensional submanifold of a high-dimensional manifold. This technique and perspective are proposed for the first time, making the analysis of synthetic data from non-convex optimization become possible.

The risk of linkability has been theoretically and practically demonstrated for a wide range of data types: tabular micro-level datasets [Narayanan and Shmatikov, 2008, Sweeney, 2002], social graph data [Narayanan et al., 2011, Narayanan and Shmatikov, 2009], aggregate statistics [Pyrgelis et al., 2017], statistical models [Shokri et al., 2017], and black-box attacks [Stadler et al., 2022]. Linkage attacks on tabular microdata typically aim to match a target record (associated with an identity) to a single record in a sensitive database, from which direct identifiers have been removed.

Regarding attribute inference attacks, existing research has established a fairly complete framework. Most studies model this attack as the adversary using a machine learning model and incomplete information about a data point to infer the missing information for that point [Fredrikson et al., 2014, 2015, Wu et al., 2016]. In our work, we characterize the advantage of an attribute inference adversary as their ability to infer a target feature given an incomplete point from the training data, relative to their ability to do so for points from the general population.

## B  Background

### B.1  Reduced Kernel Mean Embedding

**Kernel Mean Embedding.** KME [Smola et al., 2007] is a technique in machine learning that maps the mean of a probability distribution into a Reproducing Kernel Hilbert Space (RKHS). Given a probability distribution $\mathcal{P}$ over a domain $X$, and a kernel function $k : X \times X \to \mathbb{R}$ [Schölkopf and Smola, 2002], the KME of $\mathcal{P}$ is defined as the following:

$$\mu(\mathcal{P}) = \int k(x, \cdot) d\mathcal{P}(x)$$

where $\mu(\mathcal{P})$ represents the kernel mean embedding of the distribution $\mathcal{P}$, $x$ is an element of the domain $X$, and '·' denotes a placeholder for a second argument in the kernel function. Denote the associated RKHS as $\mathcal{H}_k$ and $\phi : x \in X \to k(x, \cdot) \in \mathcal{H}_k$ the corresponding canonical feature map. The kernel function $k$ quantifies the similarity between pairs of data points and is required to be positive definite to induce a valid RKHS.

Kernel Mean Embedding (KME) exhibits an array of beneficial properties, which contribute to its appeal as a robust method for diverse machine learning endeavors, most notably in the realm of specification. By the reproductiong property, $\forall f \in \mathcal{H}_k, \langle f, \mu(\mathcal{P}) \rangle = \mathbb{E}_{\mathcal{P}}[f(X)]$, which demonstrates the notion of mean. By using characteristic kernels [Sriperumbudur et al., 2011], no information about the distribution $\mathcal{P}$ would be lost during kernel embedding, i.e. $||\mu(\mathcal{P}) - \mu(\mathcal{P}')||_{\mathbb{H}_k} = 0$ implies that $\mathcal{P} = \mathcal{P}'$. One of the most commonly used samples is the Gaussian kernel

$$k(x, x') = exp(-\gamma ||x - x'||_2^2), \gamma > 0$$

The reproducing property of the Gaussian kernel and its characteristic as a characteristic kernel make it a widely used kernel in Kernel Mean Embedding (KME) [Gretton et al., 2012, Muandet and Schölkopf, 2013, Doran, 2013]. In the learnware market, for tabular data, the RKME corresponding to the Gaussian kernel is currently in use, while for image and text data, the RKME corresponding to the Gaussian kernel can also be used after extracting embeddings [Tan et al., 2024a, 2023, Xie et al., 2023]. In the privacy proofs that follow in this paper, we mainly consider the privacy protection capabilities of the RKME induced by the Gaussian kernel.

In learning tasks , we often have no accsee to the true distribution $\mathcal{P}$, but we can use samples to estimate it. The empirical kernel mean embedding is an approximation of the KME based on a finite set of samples from a probability distribution. Given a probability distribution $\mathcal{P}$ over a domain $X$, a kernel function $k : X \times X \to \mathbb{R}$, and a set of $n$ independent and identically distributed (i.i.d.) samples $\{x_1, x_2, \ldots, x_n\}$ drawn from $P$, the empirical KME $\hat{\mu}(\mathcal{P})$ of $P$ is defined as:

$$\hat{\mu}(\mathcal{P}) = \frac{1}{n} \sum_{i=1}^{n} k(x_i, \cdot)$$

The empirical KME $\hat{\mu}$ will converge to $\mu$ in the rate of $\mathcal{O}(1/\sqrt{n})$ measured by RKHS norm $|| \cdot ||_{\mathcal{H}}$ under mild conditions [Smola et al., 2007].

**Reduced Kernel Mean Embedding.**  Although the properties of KME are desirable, the computation of KME becomes challenging when there are many samples, and the calculation of KME requires access to the original data. Therefore, KME is not a specfication for learnware.To address this issue, [Wu et al., 2023] introduce RKME to approximate original KME via the reduced set method, which is first used to speed up SVM prediction [Burges, 1996]and receives more comprehensive studies in [Scholkopf et al., 1999].

The idea of RKME is to find a set $(\beta_j, z_j)_{j=1}^{m}$ and compute $\sum_{j=1}^{m} \beta_j k(z_j, \cdot)$ to approximates the KME of original data $\{x_i\}_{i=1}^{n}$, i.e. we want to solve

$$\min_{\beta, x} || \frac{1}{n} \sum_{i=1}^{n} k(x_i, \cdot) - \sum_{j=1}^{m} \beta_j k(z_j, \cdot) ||_{\mathcal{H}} \tag{9}$$

where $\beta_j \in \mathbb{R}$ is the coefficient and $x_i \in X$ is the reduced sample. The above problem is known as the reduced set construction [Scholkopf et al., 1999], when $z_j$ is newly constructed samples. Several algorithms can be used for handling the above problem.

KME $\tilde{\mu}$ enjoys a linear convergence rate $O(e^{-m})$ to empirical KME $\hat{\mu}$ when $\mathcal{H}$ is finite dimensional, which makes it a good approximation of the distribution. Meanwhile, the raw data are inaccessible to users. In this paper, we address the problem of utilizing RKME as the specification for the learnware paradigm, aiming for the efficient retrieval and organization of learnware. Consequently, both uploaders and users of learnware are required to submit the aforementioned RKME derived from their datasets, that is, the corresponding $(\beta_j, z_j)_{j=1}^{m}$. Our goal is to demonstrate that by submitting $(\beta_j, z_j)_{j=1}^{m}$, excessive information from the original dataset is not disclosed, ensuring that such a specification can preserve the privacy of the original data.

## B.2   Synthetic data

The issue of releasing synthetic data to enable data analysis and public availability while safeguarding privacy has garnered widespread attention  [Drechsler and Reiter, 2010, Bellovin et al., 2019, Arnold and Neunhoeffer, 2020, Stadler et al., 2020, England, 2022, Stadler et al., 2022, Xu et al., 2019]. Synthetic datasets are designed to retain the statistical characteristics of the original data while eliminating personal data, thereby protecting personally identifiable information [Stadler et al., 2020]. In research on synthetic data, it's commonly held that synthetic data, being 'artificial', have no direct link to real data. This assumption leads to analyses focusing only on similarity tests between real and synthetic records to assess the privacy risks of synthetic datasets [Choi et al., 2017, Yale et al., 2019a,b].

However, Stadler et al. [2022] note that not all synthetic data can withstand traditional attacks on data, such as linkage [Elliot et al., 2018, Sweeney, 2002], and attribute disclosure [Elliot et al., 2018, Machanavajjhala et al., 2007]. Common defenses against such attacks usually rely on formal privacy guarantees during the generative model training process to prevent privacy breaches [Abowd and Vilhuber, 2008, Bindschaedler et al., 2017], or involve adding noise to the generation process of synthetic data to meet differential privacy criteria [Xin et al., 2022, Jordon et al., 2018]. However, research is still lacking on whether synthetic data produced by a deterministic generation mechanism can naturally resist these two types of attacks without the addition of extra noise. In this paper, we aim to demonstrate that the RKME specification's deterministic generation mechanism, when using

an appropriate number of synthetic points, can effectively defend against these two types of attacks, while also performing well in learnware identification.

Many studies have explored similar issues, such as non-parametric models for image generation and adversaries with white-box or query access to the model [Chen et al., 2020, Hayes et al., 2019, Reiter et al., 2014], as well as considerations where the generating mechanism is treated as a complete black box, with no allowance for querying the model [Stadler et al., 2022]. In this paper, we allow the disclosure of the RKME generation mechanism and permit arbitrary queries to the RKME generation process. It is important to note that our discussion is limited to the potential privacy leakage of specifications as synthetic datasets in the learnware market. We assume that a learnware uploader's model is not subject to querying in our discussion, as this strays from the focus on the privacy of learnware models and extends beyond the scope of privacy concerns in the learnware market.

## B.3 Notations and technical overview

| Notations | Representation |
|---|---|
| $X$ | Sample space |
| $D = \{x_i\}_{i=1}^n$ | Raw data set |
| $\mathbf{D} = (x_1, x_2, \cdots, x_n)$ | Raw data set's coordinate |
| $M$ | Manifold composed of $\mathbf{D}$ |
| $\mathcal{P}_D$ | Distribution of Raw data |
| $Z = \operatorname{argmin}_{(\beta,z)} F((\beta,z), D)$ | Generation of RKME |
| $n$ | Number of raw data |
| $m$ | Number of synthetic data |
| $d$ | Dimension of data |
| $k(\cdot, \cdot)$ | (Gaussian) kernel function |
| $\|\cdot\|_{\mathcal{H}}$ | RKHS iuduced by $k(\cdot, \cdot)$ |

In our study, we equip $\mathbb{R}^n$ with the conventional Euclidean product and norm, represented by $\langle \cdot, \cdot \rangle$ and $\|\cdot\|$, respectively. We define $B(x, r)$ as the open ball within $\mathbb{R}^n$, centered at $x$ with radius $r$. For any subspace $H \subset \mathbb{R}^n$, the notation $B_H(x, r)$ signifies the intersection of $H$ and $B(x, r)$, constituting the open ball in $H$ for the induced norm.

This paper utilizes fundamental concepts of the geometry of submanifolds in Euclidean space $\mathbb{R}^n$ for those not familiar with the subject. We consider $C^\infty$ Riemannian manifolds $(M, g)$, informally referred to as smooth, which encompass an abstract manifold $M$ equipped with a $C^\infty$ atlas and a $C^\infty$ metric $g$. Such manifolds can always be isometrically embedded into some Euclidean space, implying that the pull-back of the canonical Euclidean metric aligns with the manifold's metric. A smooth submanifold $M \subset \mathbb{R}^n$ indicates that $M$ is the image of an embedding of a smooth abstract Riemannian manifold.

In the context of comparing the smoothness of manifolds across a range of models, we adopt a canonical parametrization. Specifically, we utilize the exponential map; for any smooth submanifold $M \subset \mathbb{R}^n$ and any point $x \in M$, this map defines a smooth parametrization of $M$ around $x$. The parameter $\varepsilon$ is selected to be sufficiently small, and the maximal value of $\varepsilon$ is termed the injectivity radius at $x$, denoted as $\operatorname{inj}_M(x)$. For closed subsets $M$ of $\mathbb{R}^n$, exponential maps are defined for entire tangent spaces, as per the Hopf-Rinow theorem.

The volume measure of a submanifold $M$ of dimension $d$, denoted by $\mu_M$, is the restriction of the $d$-dimensional Hausdorff measure $\mathcal{H}^d$ to $M$. This volume measure aligns with the standard definition of volume measure in a Riemannian manifold. If $\psi : M \to \mathbb{R}$ is a continuous function with support within a certain range of $x$, the volume measure can be expressed as an integral involving the function $\psi$, the metric tensor $g^x(v)$, and a chosen orthonormal basis of $T_x M$. The volume of $M$, simply denoted as $\operatorname{vol} M$, is finite for compact submanifolds of $\mathbb{R}^n$.

**Technical Overview.** The starting point of this paper regarding the data protection capability of RKME is that the generation process of RKME compresses the data to a great extent, thereby losing a significant amount of personal data information. Additionally, due to the irrationality and nonlinearity of the chosen Gaussian kernel, the generated RKME almost contains no original individual data information. We model this mathematically as follows: given the RKME $Z$, there are many possible original datasets $D$ that could generate it. After we establish a coordinate correspondence for the

original dataset $D$ as $\mathbf{D}$, we consider the set $M_Z$ in $\mathbb{R}^n$ consisting of all points $\mathbf{D}$ that could generate the RKME $Z$.

We first attempt to prove that $M_Z$ forms a manifold in $\mathbb{R}^n$. This proof will allow us to use various geometric tools for analysis, and demonstrating that such a set, constructed from the solutions to nonlinear equations, is a manifold is not trivial. This represents one of the most challenging parts of the upcoming proof.

After proving that $M_Z$ constitutes a manifold, we aim to show that the tangent space of this manifold does not contain points aligned with the coordinate axes, indicating that these points form a null measure set. This can be better understood through Ricci curvature: we are essentially proving that at almost every point on this manifold, the Ricci curvature is non-zero. The significance of this condition is that we consider those parts of the manifold $M_Z$ that expose the original data to be a lower-dimensional submanifold, thus constituting a null measure set.

Finally, we need to analyze the measure of this null set if we take a neighborhood of radius $\delta$ around each point. Since $\delta$ is relatively small, this is equivalent to determining the measure of the lower-dimensional submanifold in the corresponding dimensional space relative to the measure of $M_Z$ itself. This quantity can be interpreted as the risk level of RKME potentially exposing individual privacy. In the subsequent proofs, we will follow this flow of reasoning.

## C Proofs

### C.1 Lemmas

Recall the generation mechanism of RKME

$$F\left(x_1, \cdots, x_n; \beta_1, \cdots, \beta_m; z_1, \cdots, z_m\right) = \left\| \frac{1}{n} \sum_{i=1}^{n} k\left(x_i, \cdot\right) - \sum_{j=1}^{m} \beta_j k\left(z_j, \cdot\right) \right\|_{\mathcal{H}} \tag{10}$$

Upon substituting $k(\cdot, \cdot)$ with the Gaussian kernel and conducting an expansion, we derive the following expression,

$$
\begin{aligned}
F^2 &= \left\| \frac{1}{n} \sum_{i=1}^{n} k\left(x_i, \cdot\right) - \sum_{j=1}^{m} \beta_j k\left(z_j, \cdot\right) \right\|_{\mathcal{H}} \\
&= \left\langle \frac{1}{n} \sum_{i=1}^{n} k\left(x_i, \cdot\right) - \sum_{j=1}^{m} \beta_j k\left(z_j, \cdot\right), \frac{1}{n} \sum_{i=1}^{n} k\left(x_i, \cdot\right) - \sum_{j=1}^{m} \beta_j k\left(z_j, \cdot\right) \right\rangle_{\mathcal{H}} \\
&= \left\langle \frac{1}{n} \sum_{i=1}^{n} k\left(x_i, \cdot\right), \frac{1}{n} \sum_{i=1}^{n} k\left(x_i, \cdot\right) \right\rangle_{\mathcal{H}} - 2 \left\langle \frac{1}{n} \sum_{i=1}^{n} k\left(x_i, \cdot\right), \sum_{j=1}^{m} \beta_j k\left(z_j, \cdot\right) \right\rangle_{\mathcal{H}} \\
&\quad + \left\langle \sum_{j=1}^{m} \beta_j k\left(z_j, \cdot\right), \sum_{j=1}^{m} \beta_j k\left(z_j, \cdot\right) \right\rangle_{\mathcal{H}} \\
&= \frac{1}{n^2} \sum_{i=1}^{n} \sum_{j=1}^{n} k\left(x_i, x_j\right) - \frac{2}{n} \sum_{i=1}^{n} \sum_{j=1}^{m} \beta_j k\left(x_i, z_j\right) + \sum_{i=1}^{m} \sum_{j=1}^{m} \beta_i \beta_j k\left(z_i, z_j\right) \\
&= \frac{1}{n^2} \sum_{i=1}^{n} \sum_{j=1}^{n} e^{-\gamma(x_i - x_j)^2} - \frac{2}{n} \sum_{i=1}^{n} \sum_{j=1}^{m} \beta_j e^{-\gamma(x_i - z_j)^2} + \sum_{i=1}^{m} \sum_{j=1}^{m} \beta_i \beta_j e^{-\gamma(z_i - z_j)^2}
\end{aligned}
\tag{11}
$$

To infer the properties of $F$ by analyzing those of $F^2$, we consider this approach valid due to the following reason: given a set of samples $\{x_i\}_{i=1}^{n}$, treating $F$ and $F^2$ as functions of $(\beta_j, z_j)$, it can be shown that $F^2$ and $F$ share the same critical set, and the points where they attain their minimum

values coincide. This conclusion is readily provable as $F$ is generally considered to be non-zero in our discussions.

Let us denote $F^2$ as $G$. Now, considering the first derivatives of $G$, we have:

$$\frac{\partial G}{\partial \beta_i} = -\frac{2}{n} \sum_{k=1}^{n} e^{-\gamma(x_k - z_i)^2} + \sum_{k=1}^{m} \beta_j e^{-\gamma(z_k - z_i)^2} + \beta_i$$

$$\frac{\partial G}{\partial z_i} = -\frac{2\beta_i}{n} \sum_{k=1}^{n} 2\gamma (x_k - z_i) e^{-\gamma(z_i - x_k)^2} + \beta_i \sum_{k=1}^{m} \beta_j 2\gamma (z_i - z_k) e^{-\gamma(z_k - z_i)^2}$$

For the second derivatives, we obtain:

$$\frac{\partial^2 G}{\partial \beta_i \partial \beta_j} = \sum_{i=1}^{m} \beta_j e^{-\gamma(z_j - z_i)^2} + \delta_{ij}$$

$$\frac{\partial^2 G}{\partial z_i \partial z_j} = -\frac{2\beta_i \delta_{ij}}{n} \sum_{k=1}^{n} \left( 4\gamma^2 (x_k - z_i)^2 - 2\gamma \right) e^{-\gamma(x_k - z_i)^2}$$
$$+ (1 - \delta_{ij}) \beta_i \beta_j \left( 4\gamma^2 (z_i - z_j)^2 - 2\gamma \right) e^{-\gamma(z_j - z_i)^2}$$

This analysis serves as a foundational step in understanding the underlying properties of the RKME specification, particularly in the context of Gaussian kernels and their influence on the optimization landscape of the RKME formulation.

Next, we consider the relationship between $\mathbf{D}$ and $\mathbf{Z}$. We begin by the following lemmas:

**Lemma C.1.** *The* Reproducing kernel Hilbert space *RKHS of Gaussian kernel is separable.*

*Proof.* Since $\mathbb{R}^n$ is separable, we can take any countable dense subset $S$ of $\mathbb{R}^n$. Given that the Gaussian kernel is continuous, the set $\{k(x, \cdot) | x \in S\}$ forms a dense subset in $\mathcal{H}$. $\square$

**Lemma C.2.** *Given a fixed $\mathbf{D}$, if the corresponding equation, Eq. 1, does not equal zero, i.e.,*

$$\left\| \frac{1}{n} \sum_{i=1}^{n} k(x_i, \cdot) - \sum_{j=1}^{m} \beta_j k(z_j, \cdot) \right\|_{\mathcal{H}} > 0 \tag{12}$$

*then $\mathbf{D}$ has a unique corresponding $\mathbf{Z}$.*

*Proof.* Based on Lemma C.1, we understand that $\mathcal{H}$ is isomorphic to $l_2$. Once we fix $m$, if Eq. 1 is not zero, it implies that $\frac{1}{n} \sum_{i=1}^{n} k(x_i, \cdot)$ does not lie in any $m$-dimensional subspace of $\mathcal{H}$. Suppose the optimal set of RKME solutions corresponding to it is $\mathbf{Z} = \{z_1, \ldots, z_m\}$. Next, we will demonstrate its uniqueness. Let us denote $R = \left\| \sum_{j=1}^{m} \beta_j k(z_j, \cdot) \right\|_{\mathcal{H}}$.

**Step 1** In the first step, we prove that $\{z_1, \ldots, z_m\}$ are linearly independent and form a basis of an $m$-dimensional subspace of $\mathcal{H}$. Suppose, for the sake of contradiction, that they are linearly dependent; then the subspace spanned by $\{z_1, \ldots, z_m\}$ is less than $m$-dimensional. Therefore, we can select $S = \{s_1, \ldots, s_{m-1}, s_m, \ldots\}$ such that $\{k(s_i, \cdot)\}$ forms a countable basis of $\mathcal{H}$. Since the subspace spanned by $\{z_1, \ldots, z_m\}$ is at most $(m-1)$-dimensional, without loss of generality, we let the subspace spanned by $\{z_1, \ldots, z_m\}$ be a subspace of the space spanned by $s_1, \ldots, s_{m-1}$.

Now, given $D$, we can rewrite the empirical KME of $D$, $\frac{1}{n} \sum_{i=1}^{n} k(x_i, \cdot)$, in terms of the basis $S$:

$$\frac{1}{n} \sum_{i=1}^{n} k(x_i, \cdot) = \sum_{i=1}^{\infty} \alpha_i k(s_i, \cdot) \tag{13}$$

where $\sum_{i=1}^{\infty} |\alpha_i| < \infty$, and among the $\alpha_i$, only a number greater than $m$ and less than $n$ are nonzero. Without loss of generality, we assume that the $\alpha_i$ are arranged in decreasing order.

At this point, the optimal projection of $\frac{1}{n} \sum_{i=1}^{n} k(x_i, \cdot)$ onto the $m-1$-dimensional subspace is $\sum_{i=1}^{m-1} \alpha_i k(s_i, \cdot)$, because

$$\left\| \frac{1}{n} \sum_{i=1}^{n} k(x_i, \cdot) - \sum_{i=1}^{m} \alpha_i k(s_i, \cdot) \right\|_{\mathcal{H}} \tag{14}$$

achieves the minimal value. We denote this optimal projection as $R'$. Now we estimate $R - R'$:

$$R - R' \geq \|\alpha_m k(s_m, \cdot)\|_{\mathcal{H}} > 0 \tag{15}$$

This leads to a contradiction.

**Step 2** Let's consider the set $S = \{z_1, \ldots, z_m, s_{m+1}, s_{m+2}, \ldots\}$, ensuring that $\{k(z_1, \cdot), \ldots, k(z_m, \cdot), k(s_{m+1}, \cdot), \ldots\}$ forms a countable basis in $\mathcal{H}$. Suppose there exists another set $\{y_1, \ldots, y_m\}$ that achieves the same value for Eq. 1 as $\{z_1, \ldots, z_m\}$ with respect to $\frac{1}{n} \sum_{i=1}^{n} k(x_i, \cdot)$. In that case, we can expand $\{y_1, \ldots, y_m\}$ with the basis $\{s_{m+1}, s_{m+2}, \ldots\}$ to form another basis of $\mathcal{H}$, denoted as $S' = \{k(y_1, \cdot), \ldots, k(y_m, \cdot), k(s'_{m+1}, \cdot), \ldots\}$.

Similar to the proof in Step 1, we know that the optimal $m$-dimensional projection subspaces of the empirical KME $\frac{1}{n} \sum_{i=1}^{n} k(x_i, \cdot)$ onto the two bases are generated by $\{z_1, \ldots, z_m\}$ and $\{y_1, \ldots, y_m\}$, respectively. Next, we will prove that the elements in $\{z_1, \ldots, z_m\}$ and $\{y_1, \ldots, y_m\}$ correspond to each other and are equal. Let us denote the optimal $m$-dimensional projection subspace of $\frac{1}{n} \sum_{i=1}^{n} k(x_i, \cdot)$ onto both bases as $\mathcal{H}'$.

To prove this, we first establish two auxiliary propositions:

**Proposition C.3.** *The Gaussian kernel is a strictly positive-definite kernel.*

*Proof.* Since the Fourier transform of a Gaussian function is also a Gaussian function and always positive, the proposition holds. $\qquad\square$

**Proposition C.4.** *For $\{x_1, \ldots, x_n\}$, the following two statements are equivalent:*

1. *$\{k(x_1, \cdot), \ldots, k(x_n, \cdot)\}$ are linearly independent.*

2. *The points $\{x_1, \ldots, x_n\}$ are all distinct.*

*Proof.* It is obvious that [1] implies [2]; we only need to prove that [2] implies [1]. Assume that the points $\{x_1, \ldots, x_n\}$ are all distinct. Suppose there exist real numbers $c_1, c_2, \ldots, c_n$ such that

$$\sum_{i=1}^{n} c_i k(x_i, \cdot) = 0.$$

Recall that in an RKHS, the inner product is defined as

$$\langle f, g \rangle_{\mathcal{H}} = \sum_{i=1}^{m} \sum_{j=1}^{l} \alpha_i \beta_j k(x_i, x'_j),$$

$$\text{where } f = \sum_{i=1}^{\infty} \alpha_i k(x_i, \cdot), \quad g = \sum_{j=1}^{\infty} \beta_j k(x'_j, \cdot). \tag{16}$$

Based on our assumption, we compute

$$\left\langle \sum_{i=1}^{n} c_i k(x_i, \cdot), \sum_{j=1}^{n} c_j k(x_j, \cdot) \right\rangle_{\mathcal{H}} = 0.$$

Using the linearity of the inner product, we expand this as

$$\sum_{i=1}^{n} \sum_{j=1}^{n} c_i c_j k(x_i, x_j) = 0.$$

Let $K$ be the $n \times n$ symmetric kernel matrix with elements

$$K_{ij} = k(x_i, x_j) = \exp\left(-\frac{\|x_i - x_j\|^2}{2\sigma^2}\right).$$

Then the above equation can be written as

$$c^\top K c = 0.$$

Since the Gaussian kernel is strictly positive-definite, $K$ is positive-definite, implying that $c^\top K c > 0$ unless $c = 0$. Therefore, we must have $c = 0$, which contradicts the assumption that there exist nonzero $c_i$. Hence, the proposition is proved. $\qquad\square$

Using Proposition C.4, for the sets $\{z_1, \ldots, z_m\}$ and $\{y_1, \ldots, y_m\}$, we know that the functions $\{k(z_1, \cdot), \ldots, k(z_m, \cdot), k(y_1, \cdot), \ldots, k(y_m, \cdot)\}$ are linearly dependent. Therefore, there exist $p, q \in \{1, 2, \ldots, m\}$ such that $z_p = y_q$. Without loss of generality, let $z_p = z_m$ and $y_q = y_m$. The remaining elements $\{z_1, \ldots, z_{m-1}\}$ and $\{y_1, \ldots, y_{m-1}\}$ are still linearly dependent. By mathematical induction, we thus complete the proof.

$\qquad\square$

**Lemma C.5.** *The set of all possible* $\mathbf{D}$ *that make the corresponding equation Eq. 1 equal to zero has measure zero in $\mathbb{R}^n$. That is,*

$$\mu\left\{(x_1, \ldots, x_n) \in \mathbb{R}^n \mid \exists\{\beta_1, \cdots, \beta_m, z_1, \cdots, z_m\}, s.t. \left\|\frac{1}{n}\sum_{i=1}^n k(x_i, \cdot) - \sum_{j=1}^m \beta_j k(z_j, \cdot)\right\|_{\mathcal{H}} = 0\right\} = 0 \tag{17}$$

*Proof.* We say that any $f \in \mathcal{H}$ can be represented by $t$ elements in $\mathbb{R}$ via the kernel $k(\cdot, \cdot)$ if there exist $(s_1, \ldots, s_t)$ such that

$$\left\|f - \sum_{j=1}^t \beta_j k(s_j, \cdot)\right\|_{\mathcal{H}} = 0. \tag{18}$$

We define the representational dimension of $f$ as the infimum of such $t$, denoted as $t_f$.

Similar to the proof idea in Lemma C.2, we know that the representational dimension $t_k \leq m$ for the empirical KME $\frac{1}{n}\sum_{i=1}^n k(x_i, \cdot)$.

For $t_k = c$ and the set $\{x_1, \ldots, x_n\}$, from Proposition C.4, we easily obtain that there exist $p, q \in \{1, 2, \ldots, n\}$ such that $x_p = x_q$.

Therefore, all $\{x_1, \ldots, x_n\}$ with $t_k = c$ are formed by the union of at most $\binom{n}{c}$ $c$-dimensional subspaces; denote this union as $\mathcal{H}'_c$. By dimensional considerations, it is easy to see that $\mathcal{H}'_c$ has measure zero in $\mathcal{H}$.

Since a countable union of measure zero sets is still a measure zero set, we thus obtain that $\bigcup_{t=1}^\infty \mathcal{H}'_t$ still has measure zero in $\mathcal{H}$, hence the conclusion follows.

$\qquad\square$

**Lemma C.6.** *Let us fix $Z$. Denote $C \subset \mathbb{R}^n$ as the critical set of $G(\mathbf{x}, \beta, \mathbf{z})$, we have that the Lebesgue measure of $G(C)$ is $0$ in $\mathbb{R}$.*

*Proof.* We delineate a general scenario where $f : \mathbb{R}^m \to \mathbb{R}^n$ manifests as a smooth mapping. Without loss of generality, let $M = \mathbb{R}^m$ and $N = \mathbb{R}^n$. We employ mathematical induction with respect to $m$. The base case where $m = 0$ is trivially evident. Let $C$ represent the entirety of critical values of $f$, denoted as the critical set. It suffices to demonstrate that for any $y \in C$, there exists an open neighborhood around $y$ such that its intersection with $C$ constitutes a null set. Define

$$C_s = \{x \in \mathbb{R}^n \mid f \text{ has all its } k\text{-th partial derivatives equal to zero at } x, \ 1 \leq k \leq s.\}$$

Clearly, $C \supset C_1 \supset C_2 \supset \cdots$ forms a sequence of closed sets. The objective is to ascertain that $f(C_s - C_{s+1})$ are all null sets.

1. $f(C - C_1)$ is a null set. Indeed, suppose $x_0 \in C, x_0 \notin C_1$, then $f$ at $x_0$ possesses non-vanishing first-order partial derivatives, without loss of generality, let $\frac{\partial f_1}{\partial x^1} \neq 0$. Consider the mapping $g_0 : \mathbb{R}^m \to \mathbb{R}^m$, defined as

$$g_0(x) = \left(f_1(x), x^2, x^3, \cdots, x^m\right),$$

where $g_0$ at the vicinity of $x_0$ has a rank of $m$. By the inverse function theorem, open neighborhoods $U$ and $V$ around $x_0$ exist such that the restriction $g_0|_U : U \to V$ is a diffeomorphism, with its inverse denoted as $h_0$. Then,

$$f \circ h_0(x) = \left(x^1, f_2 \circ h_0(x), \cdots, f_m \circ h(x)\right),$$

and $f(C \cap h_0(V)) = f \circ h_0(h_0^{-1}(C) \cap V)$. If we define

$$k_t\left(x^2, x^3, \cdots, x^m\right) = \left(f_2 \circ h_0\left(t, x^2, \cdots, x^m\right), \cdots, f_m \circ h_0\left(t, x^2, \cdots, x^m\right)\right),$$

then,

$$h_0^{-1}(C) \cap V = \bigcup_t \{t\} \times \mathrm{Crit}(k_t),$$

where $\mathrm{Crit}(k_t)$ signifies the critical points of $k_t$. By the induction hypothesis, $k_t(\mathrm{Crit}(k_t))$ are null sets in $\mathbb{R}^{m-1}$, hence,

$$f(C \cap h_0(V)) = \bigcup_t \{t\} \times k_t(\mathrm{Crit}(k_t))$$

is a null set in $\mathbb{R}^n$.

2. $f(C_s - C_{s+1})$ is a null set. Let $x_0 \in C_s - C_{s+1}$, and assume without loss of generality that $\frac{\partial f'}{\partial x^1}(x_0) \neq 0$, where

$$f' = \frac{\partial^{i_1 + \cdots + i_m} f}{\partial \left(x^1\right)^{i_1} \cdots \partial \left(x^m\right)^{i_m}}, \quad i_1 + \cdots + i_m = s.$$

Similarly as before, consider the mapping $g_s : \mathbb{R}^m \to \mathbb{R}^m$, defined by

$$g_s(x) = \left(f'(x), x^2, \cdots, x^m\right),$$

where $g_s$ near $x_0$ functions as a diffeomorphism, with its inverse denoted as $h_s : V \to \mathbb{R}^m$. Let $k_s = f \circ h_s$ and denote

$$k' = k_s|_{\{0\} \times \mathbb{R}^{m-1} \cap V}.$$

It is evident that $g_s(C_s \cap h_s(V)) \subset \{0\} \times \mathbb{R}^{m-1} \cap V$, and

$$f(C_s \cap h_s(V)) \subset k'(\mathrm{Crit}(k')),$$

from which, by the inductive assumption, $f(C_s \cap V)$ is a null set.

3. For sufficiently large $s$, $f(C_s)$ is a null set. Suppose $x_0 \in C_s$ with $s > \frac{m}{n} - 1$. Select a cube $I$ centered at $x_0$ with side length $\delta$. By the Taylor expansion of multivariate functions, a constant $M > 0$ exists such that

$$\|f(x) - f(y)\| \leq M\|x - y\|^{s+1}, \quad \forall x \in C_s \cap I, y \in I.$$

Subdivide $I$ into $N^m$ smaller cubes with side length $\frac{\delta}{N}$. If $I'$ is one of the subdivided cubes, then the aforementioned inequality implies that when $I' \cap C_s \neq \emptyset$, $f(I')$ is contained within a cube of side length not exceeding

$$2M\sqrt{m}\left(\frac{\delta}{N}\right)^{s+1},$$

hence $f(C_s \cap I)$ is enveloped within a union of cubes whose total volume does not exceed

$$N^m \cdot \left[2M\sqrt{m}\left(\frac{\delta}{N}\right)^{s+1}\right]^n = \left[2M\sqrt{m}\delta^{s+1}\right]^n \cdot N^{m-n(s+1)}.$$

For $s > \frac{m}{n} - 1$, choosing $N$ sufficiently large ensures that this volume becomes arbitrarily small. Thus, $f(C_s \cap I)$ is a null set, and consequently, $f(C_s)$ is a null set.

Therefore, we conclude that $f(C)$ is a null set in $\mathbb{R}^n$. $\qquad\square$

Now we revisit the definition of $G$ in Eq. 11. Our central idea in the proof is to fix $\{\beta_i, z_i\}, i = 1, 2, \ldots, m$, so that $G(x_1, \ldots, x_n)$ becomes a function on $\mathbb{R}^n$. From Lemma C.1, we know that a fixed set $\{x_1, \ldots, x_n\}$ usually corresponds to a set $\{\beta_1, \ldots, \beta_n, z_1, \ldots, z_n\}$. However, when we fix a $\{\beta_1, \ldots, \beta_n, z_1, \ldots, z_n\}$, the possible $\{x_1, \ldots, x_n\}$ that can generate this RKME are not unique and may even form a manifold in $\mathbb{R}^n$.

Therefore, following the notation in the main text, we denote the $\{x_1, \ldots, x_n\}$ contained in a sample set $D$ as a point $\mathbf{D} = (x_1, \ldots, x_n)$ in $\mathbb{R}^n$. Next, we will prove that all such $\mathbf{D} = (x_1, \ldots, x_n)$ corresponding to a given $\{\beta_1, \ldots, \beta_n, z_1, \ldots, z_n\}$ form a manifold. To this end, we define the following mapping $f$ from $\mathbb{R}^n$ to $\mathbb{R}^{2m}$:

$$
\begin{aligned}
f(x_1, x_2, \cdots, x_n) &= (\tfrac{\partial G}{\partial z_1}, \cdots, \tfrac{\partial G}{\partial z_m}, \tfrac{\partial G}{\partial \beta_1}, \cdots, \cdots, \tfrac{\partial G}{\partial \beta_m}) \\
&= (-\tfrac{2}{n} \sum_{k=1}^n e^{-\gamma(x_k - z_1)^2} + \sum_{k=1}^m \beta_j e^{-\gamma(z_k - z_1)^2} + \beta_1, \cdots, \\
&\cdots, -\tfrac{2\beta_i}{n} \sum_{k=1}^n 2\gamma (x_k - z_m) e^{-\gamma(z_m - x_j)^2} + \beta_n \sum_{k=1}^m \beta_j 2\gamma (z_m - z_j) e^{-\gamma(z_k - z_m)^2}.
\end{aligned}
\tag{19}
$$

We now consider the total differential $df(x)$ of the function $f$ defined in Eq. 19, which is the Jacobian $Jf(x)$. To do this, we first calculate the following equation:

$$
\begin{aligned}
\frac{\partial f}{\partial x_i} = (&\frac{4\gamma(x_i - z_1)}{n} e^{-\gamma(x_i - z_1)^2}, \cdots, \frac{4\gamma(x_i - z_m)}{n} e^{-\gamma(x_i - z_m)^2}, \\
&\frac{8\gamma\beta_1(x_i - z_1)^2 - 4\gamma\beta_1}{n} \sum_{k=1}^n e^{-\gamma(x_i - z_1)^2}, \cdots, \frac{8\gamma\beta_m(x_i - z_m)^2 - 4\gamma\beta_m}{n} \sum_{k=1}^n e^{-\gamma(x_i - z_m)^2}),
\end{aligned}
$$

and $Jf(x)$ is $\begin{bmatrix} \frac{\partial f_1}{\partial x_1} & \cdots & \frac{\partial f_1}{\partial x_n} \\ \vdots & \ddots & \vdots \\ \frac{\partial f_{2m}}{\partial x_1} & \cdots & \frac{\partial f_{2m}}{\partial x_n} \end{bmatrix}$, and thus can be writen as $\begin{bmatrix} \frac{\partial f}{\partial x_1} & \cdots & \frac{\partial f}{\partial x_n} \end{bmatrix}$. Next, regarding the analysis of the properties of $Jf(x)$, we first prove a lemma in general case:

**Lemma C.7.** *For the function $f : M^n \to N^n$ is $C^k$ mapping and $rank_p f = n$, then there exists an open neighborhood $U$ in $\mathbb{R}^n$ and an open neighborhood $V$ of $q = f(p)$ such that the restriction $f|_U : U \to V$ is a diffeomorphism.*

*Proof.* By composing with an invertible linear map, we may also start from that the Jacobian of $f$ at the origin is the identity matrix, i.e., $Jf(0) = I_n$. In this case, near the origin, $f$ is a small perturbation of the identity mapping, which can be defined as the perturbation term

$$
g : \mathbb{R}^n \to \mathbb{R}^n, \quad g(x) = f(x) - x, \quad x \in \mathbb{R}^n.
$$

Since $Jg(0) = 0$, there exists $\epsilon > 0$ such that

$$
\|Jg(x)\| \leqslant \frac{1}{2}, \quad \forall x \in \overline{B_\epsilon(0)}.
$$

From the mean value theorem for multivariate vector-valued functions, we have

$$
\|g(x_1) - g(x_2)\| \leqslant \|Jg(\xi)\| \|x_1 - x_2\| \leqslant \frac{1}{2} \|x_1 - x_2\|, \quad \forall x_1, x_2 \in \overline{B_\epsilon(0)}.
$$

Let $y \in B_{\frac{\epsilon}{2}}(0)$ and consider solving the equation

$$
f(x) = y, \quad x \in B_\epsilon(0).
$$

This is equivalent to finding a fixed point in $B_\epsilon(0)$ for $g_y(x) = x + y - f(x)$. We use the contraction mapping principle to find this fixed point. First, we have

$$
\|g_y(x)\| \leqslant \|y\| + \|g(x)\| < \frac{\epsilon}{2} + \frac{1}{2} \|x\| \leqslant \epsilon, \quad \forall x \in \overline{B_\epsilon(0)}.
$$

This shows that $g_y(\overline{B_\epsilon(0)}) \subset B_\epsilon(0)$. The mapping $g_y : \overline{B_\epsilon(0)} \to B_\epsilon(0) \subset \overline{B_\epsilon(0)}$ is a contraction:

$$\|g_y(x_1) - g_y(x_2)\| = \|g(x_2) - g(x_1)\| \leqslant \frac{1}{2}\|x_1 - x_2\|, \quad \forall x_1, x_2 \in \overline{B_\epsilon(0)}.$$

Therefore, the equation above has a unique solution in $\bar{B}_\epsilon(0)$, denoted as $x_y$. And we know that $x_y \in B_\epsilon(0)$. Let $U = f^{-1}\left(B_{\frac{\epsilon}{2}}(0)\right) \cap B_\epsilon(0)$, $V = B_{\frac{\epsilon}{2}}(0)$. Then the above discussion indicates that $f|_U : U \to V$ is a one-to-one $C^k$ mapping, whose inverse $h(y) = x_y$ satisfies the equation

$$y - g(h(y)) = h(y).$$

We have: (1) $h : V \to U$ is continuous: when $y_1, y_2 \in V$,

$$\|h(y_1) - h(y_2)\| \leqslant \|y_1 - y_2\| + \|g(h(y_1)) - g(h(y_2))\|$$
$$\leqslant \|y_1 - y_2\| + \frac{1}{2}\|h(y_1) - h(y_2)\|.$$

Therefore, we have $\|h(y_1) - h(y_2)\| \leqslant 2\|y_1 - y_2\|$, meaning that $h$ is a Lipschitz continuous mapping.

(2) $h : V \to U$ is a differentiable mapping: Let $y_0 \in V$, then for $y \in V$, we have

$$h(y) - h(y_0) = (y - y_0) - [g(h(y)) - g(h(y_0))]$$
$$= (y - y_0) - Jg(h(y_0)) \cdot (h(y) - h(y_0)) + o(\|h(y) - h(y_0)\|).$$

From (1), we obtain

$$[I_n + Jg(h(y_0))] \cdot (h(y) - h(y_0)) = (y - y_0) + o(\|y - y_0\|),$$

thus

$$h(y) - h(y_0) = [I_n + Jg(h(y_0))]^{-1} \cdot (y - y_0) + o(\|y - y_0\|).$$

(3) $h : V \to U$ is a $C^k$ mapping. In fact, from (2), we know

$$Jh(y) = [I_n + Jg(h(y_0))]^{-1} = [Jf(h(y))]^{-1}, \quad \forall y \in V.$$

Since $f$ is a $C^k$ mapping, and from the above formula, we can successively increase the differentiability of $h$, ultimately concluding that $h$ is a $C^k$ mapping. $\qquad \square$

And the Lemma we proved above has a natural corollary:

**Corollary C.8.** *For $f$ we defined in Eq. 19, when $Jf(x)$ is a.e. of full row rank, there exists a neighborhood $U$ around $\mathbf{D}$ such that $F^{-1}(F(p)) \cap U$ forms a smooth submanifold of $M$ with dimension $n - 2m$.*

This corollary provides us with a local conclusion, namely that when $Jf(x)$ is of full row rank, fixing RKME $Z$ corresponds to a certain $\mathbf{D}$ that locally forms an $n - 2m$-dimensional manifold within its neighborhood. To study the specific correspondences between $Z$ and $D$, we still need a global conclusion. Before using Lemma 7 to prove the global result, we need to explore $Jf(x)$ further. One of the initial questions to address is whether $Jf(x)$ is of full row rank at all points. An obvious observation is that when all $x_i$ are equal, the rank of $Jf(x)$ is 1; however, such points are measure zero in $\mathbb{R}^n$. A natural question arises: Are the points where $Jf(x)$ is not of full row rank also measure zero in $\mathbb{R}^n$? To answer this question, we prove the following proposition:

**Proposition C.9.** *The Jacobian $Jf(x)$ is a.e. of full row rank.*

*Proof.* Our proof will be divided into the following three parts: we will partition $Jf(x)$ into two blocks consisting of the first $m$ rows and the last $m$ rows. First, we will prove that the $m$ row vectors in each of these two blocks are linearly independent, and then we will show that the vectors in these two blocks are also linearly independent of each other.

Step 1: We first define

$$g(x_1, x_2, \cdots, x_n) = \left(\frac{\partial G}{\partial z_1}, \cdots, \frac{\partial G}{\partial z_m}\right)$$
$$= \left(-\frac{2}{n}\sum_{k=1}^{n} e^{-\gamma(x_k - z_1)^2} + \sum_{j=1}^{m} \beta_j e^{-\gamma(z_j - z_1)^2} + \beta_1, \cdots, \right. \tag{20}$$
$$\left. -\frac{2}{n}\sum_{k=1}^{n} e^{-\gamma(x_k - z_m)^2} + \sum_{j=1}^{m} \beta_j e^{-\gamma(z_j - z_m)^2} + \beta_m\right).$$

Thus, we have $\frac{\partial g}{\partial x_i} = \left(\frac{4\gamma(x_i - z_1)}{n}e^{-\gamma(x_i - z_1)^2}, \cdots, \frac{4\gamma(x_i - z_m)}{n}e^{-\gamma(x_i - z_m)^2}\right)$ and $Jg_1(x) =$ $\begin{bmatrix} \frac{\partial g_1}{\partial x_1} & \cdots & \frac{\partial g_1}{\partial x_n} \\ \vdots & \ddots & \vdots \\ \frac{\partial f_m}{\partial x_1} & \cdots & \frac{\partial g_m}{\partial x_n} \end{bmatrix}$ . We aim to prove the following conclusion: the points where the rank of $Jg_1(x)$ is less than $m$ are measure zero in $\mathbb{R}^n$. We note the fact that the rank of $Jg_1(x)$ being less than $m$ is equivalent to the determinant of any $m$-dimensional submatrix of $Jg_1(x)$ being zero. This is evident because if the rank of $Jg_1(x)$ is less than $m$, then the first $m$ rows of $Jg_1(x)$ must be linearly dependent, thus any $m$-dimensional submatrix formed by the first $m$ rows is also linearly dependent, resulting in a determinant of zero. Conversely, the same reasoning applies.

Let us analyze an arbitrary $m$-dimensional submatrix. Without loss of generality, we can select the first $m$ column vectors. Thus, we have the matrix $J = \begin{bmatrix} \frac{\partial g_1}{\partial x_1} & \cdots & \frac{\partial g_1}{\partial x_m} \\ \vdots & \ddots & \vdots \\ \frac{\partial g_m}{\partial x_1} & \cdots & \frac{\partial g_m}{\partial x_m} \end{bmatrix}$ . Notice that $\det(J)$ is a function defined from $\mathbb{R}^m$ to $\mathbb{R}$. We want to prove that its preimage at the point $0$ is a zero measure set. According to Lemma C.6 and C.7, it suffices to show that the derivative of $\det(J)$ is almost everywhere non-zero. If this is true, then the preimage at $0$ will form a $\mathbb{R}^{m-1}$-dimensional manifold in $\mathbb{R}^m$, and thus it will be a zero measure set. In fact, if we take the derivative of $\det(J)$, we have

$$\frac{\partial \det(J)}{x_i} = \sum_{j=1}^{m} \det \begin{bmatrix} \frac{\partial g_1}{\partial x_1} & \cdots & \frac{\partial g_1}{\partial x_m} \\ \vdots & \ddots & \vdots \\ \frac{\partial^2 g_j}{\partial x_1 x_i} & \cdots & \frac{\partial^2 g_j}{\partial x_m x_i} \\ \vdots & \ddots & \vdots \\ \frac{\partial g_m}{\partial x_1} & \cdots & \frac{\partial g_m}{\partial x_m} \end{bmatrix}. \tag{21}$$

and $d\det(J) = \left(\frac{\partial \det(J)}{x_1}, \cdots, \frac{\partial \det(J)}{x_m}\right)$.

We only need to prove that the vector in $d\det(J)$ will not be simultaneously zero. Since $g$ is a smooth function, and the determinant is also a smooth function, by Lemma C.7, we know that the preimage of the case where the derivative vector is simultaneously zero is a zero-measure set.

For the last $m$ rows, we can define the function

$$h(x_1, x_2, \cdots, x_n) = \left(\frac{\partial G}{\partial \beta_1}, \frac{\partial G}{\partial \beta_2}, \cdots, \frac{\partial G}{\partial \beta_m}\right)$$
$$= \left(\frac{8\gamma\beta_1(x_1 - z_1)^2 - 4\gamma\beta_1}{n}\sum_{k=1}^{n} e^{-\gamma(x_1 - z_1)^2}, \cdots, \right. \tag{22}$$
$$\left. \frac{8\gamma\beta_m(x_n - z_m)^2 - 4\gamma\beta_m}{n}\sum_{k=1}^{n} e^{-\gamma(x_n - z_m)^2}\right).$$

The function $h$ we defined is still smooth, allowing us to draw the same conclusion. To combine the first $m$ rows and the last $m$ rows, we simply define the function $f = g \times h$, and this immediately leads us to the conclusion.

$\square$

With the above propositions and lemmas established, we can prove the following lemma:

**Lemma C.10.** *Let $f : M^m \to N^n$ be a smooth map between differentiable manifolds. If there exists a constant $l$ such that $\text{rank}_p f = l$ for all $p \in M$, then for each fixed $q \in N$,*

$$f^{-1}(q) = \{p \in M \mid f(p) = q\}$$

*is either empty or a regular submanifold of $M$ with dimension $m - l$.*

*Proof.* Let $S = f^{-1}(q)$ and assume $S$ is not empty, so there exists $p \in S$. We will prove that there exists a local coordinate system $(U, \varphi)$ around $p$ in $M$ and a local coordinate system $(V, \psi)$ around $q$ in $N$ such that $\varphi(p) = 0 \in \mathbb{R}^m$, $\psi(q) = 0 \in \mathbb{R}^n$, $f(U) \subset V$, and the local representation of $f$ is of the form

$$\psi \circ f \circ \varphi^{-1}(x^1, x^2, \cdots, x^m) = \left(x^1, x^2, \cdots, x^l, g^{l+1}(x^1, x^2, \cdots, x^l), \cdots, g^n(x^1, x^2, \cdots, x^l)\right).$$

The proof of this equality is similar to the standard form of immersion maps. We can assume $M = \mathbb{R}^m$, $N = \mathbb{R}^n$, $p = 0 \in \mathbb{R}^m$, and $q = 0 \in \mathbb{R}^n$. The map $f$ can be represented in components as

$$f(x^1, x^2, \cdots, x^m) = \left(f_1(x^1, x^2, \cdots, x^m), \cdots, f_n(x^1, x^2, \cdots, x^m)\right).$$

By assumption, the matrix

$$\left(\frac{\partial f_i}{\partial x^j}\right)_{\substack{1 \le i \le n \\ 1 \le j \le m}}$$

has rank $l$. By rearranging the order of coordinates, we can assume that the matrix

$$\left(\frac{\partial f_i}{\partial x^j}\right)_{\substack{1 \le i \le l \\ 1 \le j \le l}}$$

is non-degenerate at the origin. We define the map $g : \mathbb{R}^m \to \mathbb{R}^m$ as

$$g(x^1, x^2, \cdots, x^m) = \left(f_1, f_2, \cdots, f_l, x^{l+1}, \cdots, x^m\right).$$

The Jacobian of $g$ at the origin is of the form

$$\begin{bmatrix} \left(\frac{\partial f_i}{\partial x^j}\right)_{l \times l} & * \\ 0 & I_{m-l} \end{bmatrix}.$$

Thus, it is non-degenerate at the origin. By the Lemma C.7, there exist open neighborhoods $U$ around $0 \in \mathbb{R}^m$ and $V$ such that $g_U : U \to V$ is a diffeomorphism. We can assume $V$ is a convex neighborhood, and let $g|_U = \varphi$; thus, $\varphi$ is a local coordinate map around $p = 0$. In this local coordinate map, the local representation of $f$ is of the form

$$f \circ \varphi^{-1}(x^1, x^2, \cdots, x^m) = \left(x^1, x^2, \cdots, x^l, g^{l+1}, \cdots, g^n\right),$$

where $g^i$ for $l + 1 \le i \le n$ are functions of $(x^1, x^2, \cdots, x^l)$. Since $\operatorname{rank} J\left(f \circ \varphi^{-1}\right)\big|_V \equiv l$, we have

$$\frac{\partial g^i}{\partial x^j} = 0, \quad \forall l + 1 \le i \le n, l + 1 \le j \le m.$$

Since $V$ is a convex domain, it follows that

$$g^i = g^i(x^1, x^2, \cdots, x^l), \quad l + 1 \le i \le n.$$

Thus, we have

$$
\begin{aligned}
S \cap U = & \{s \in U \mid f(s) = 0\} \\
= & \left\{ s \in U \mid x^1(s) = \cdots = x^l(s) = 0, \right. \\
& \left. g^{l+1}(x^1(s), \cdots, x^l(s)) = \cdots = g^n(x^1(s), \cdots, x^l(s)) = 0 \right\} \\
= & \left\{ s \in U \mid x^1(s) = \cdots = x^l(s) = 0 \right\}.
\end{aligned}
$$

Finally, by taking the subtopology on $M$ and considering the local coordinates on $M \times U$ as the first $m$ components of the local coordinates on $N$ over $U$, it is straightforward to verify that $M$ is a differentiable manifold, and the inclusion map from $M$ to $N$ is an embedding. $\qquad \square$

By Prop. C.9 and Lemma C.10, we immediately obtain the following corollary:

**Corollary C.11.** *Given $Z = (\beta_1, \ldots, \beta_m, z_1, \ldots, z_m)$, the points $\mathbf{D}$ in $\mathbb{R}^n$ corresponding to all original datasets $D$ that could generate $Z$ form an $n - 2m$-dimensional manifold in $\mathbb{R}^n$.*

With Corollary C.11, we can establish the relationship between a fixed RKME $Z$ and the points $\mathbf{D}$ in $\mathbb{R}^n$ corresponding to the original dataset $D$ that may generate this RKME. Now, we can consider what we aim to prove: that the privacy of the RKME arises from data compression. Mathematically, this means that a fixed RKME $Z$ may correspond to multiple $\mathbf{D}$, and these $\mathbf{D}$ form a manifold. The significance of proving that $\mathbf{D}$ forms a manifold is that it is a set that is locally homeomorphic to Euclidean space, allowing us to use geometric methods to analyze it.

In particular, our first concern is whether an RKME $Z$ generated from a dataset $D$ contains the original points from $D$. However, based on Lemma C.2 and the fact that $\mathbf{D}$ forms a manifold, what we actually need to assess is the volume of the intersection between the sections formed by the coordinates in $Z$ and this manifold. The points in the manifold $M$ that contain RKME $Z$ correspond to the intersection points between these coordinate sections and the manifold, specifically where some component coordinates are identical.

If this manifold happens to be a plane parallel to the coordinate axes in $\mathbb{R}^n$, then the intersection with $M$ formed by sections from the coordinates in $Z$ could potentially cover the entire $M$, indicating that such a compression mechanism may not protect privacy (at least for certain datasets, it could completely disclose privacy). To characterize this, we want to determine whether there are instances on the entire manifold where a plane parallel to the coordinate axes in $\mathbb{R}^n$ exists, which can be described in mathematics using the concept of tangent spaces. Let $C^\infty(M)$ be the vector space consisting of all smooth functions on the differentiable manifold $M$. For a point $p \in M$, if a linear map $X_p : C^\infty(M) \to \mathbb{R}$ satisfies the following condition

$$
X_p(fg) = f(p)X_p g + g(p)X_p f, \quad \forall f, g \in C^\infty(M),
$$

then $X_p$ is called the tangent vector at point $p$ on $M$. The vector space formed by all tangent vectors is referred to as the tangent space at $p$, denoted as $T_p M$. Now we only need to study $T_p M$. Note that if we let $x = \varphi(q) \in \varphi(U)$ and $a = \varphi(p)$, then

$$
\begin{aligned}
f(q) = f \circ \varphi^{-1}(x) & = f \circ \varphi^{-1}(a) + \int_0^1 \left[ \frac{d}{dt} f \circ \varphi^{-1}(a + t(x - a)) \right] dt \\
& = f \circ \varphi^{-1}(a) + \sum_{i=1}^n \left( x^i - a^i \right) g_i(x),
\end{aligned}
$$

where

$$
g_i(x) = \int_0^1 \frac{\partial f \circ \varphi^{-1}}{\partial x^i}(a + t(x - a))dt.
$$

The function $g_i$ is still smooth, and

$$
g_i(a) = \frac{\partial f \circ \varphi^{-1}}{\partial x^i}(a) = \left. \frac{\partial}{\partial x^i} \right|_p f.
$$

By the definition of a tangent vector, we have

$$X_p f = X_p \left( \sum_{i=1}^{n} \left( x^i - a^i \right) g_i(x) \right) = \sum_{i=1}^{n} \left( X_p x^i \right) g_i(a) = \sum_{i=1}^{n} \left( X_p x^i \right) \left. \frac{\partial}{\partial x^i} \right|_p f,$$

thus we only need to study the derivatives of this manifold with respect to $x_i$. Therefore, we have the following lemma:

**Lemma C.12.** *Given the RKME $Z$, for the manifold $M$ formed by the points $\mathbf{D}$ that could generate $Z$, the tangent space at any point $p$ cannot be spanned by $n - 2m$ coordinate axes in $\mathbb{R}^n$.*

*Proof.* Based on the equation we mentioned earlier,

$$X_p f = X_p \left( \sum_{i=1}^{n} \left( x^i - a^i \right) g_i(x) \right) = \sum_{i=1}^{n} \left( X_p x^i \right) g_i(a) = \sum_{i=1}^{n} \left( X_p x^i \right) \left. \frac{\partial}{\partial x^i} \right|_p f,$$

we only need to prove that at any point $\mathbf{D}$ on the manifold $M$, the tangent space is spanned by at least $n - 2m + 1$ coordinate axes. This is equivalent to showing that the number of components with a derivative of zero is less than or equal to $2m - 1$.

First, we differentiate at any point $\mathbf{D}$:

$$\frac{\partial G}{\partial x_i} = -\frac{4\gamma}{n^2} \sum_{k=1,k\neq i}^{n} (x_i - x_k)e^{-\gamma(x_i - x_k)^2} + \frac{4\gamma}{n^2} \sum_{k=1}^{m} \beta_k (x_i - z_k)e^{-\gamma(x_i - z_k)^2}.$$

We can use a similar technique to that in Proposition C.9, but for this problem, there is an easier method. We take the second derivative of $\frac{\partial G}{\partial x_i}$ along a specific direction $x_j$ at any point $\mathbf{D}$, leading to the following expression:

$$\frac{\partial^2 G}{\partial x_j \partial x_i} = -\frac{2\gamma}{n^2} e^{-\gamma(x_i - x_j)^2} + \frac{4\gamma^2}{n^2} (x_i - x_j)^2 e^{-\gamma(x_i - x_j)^2}. \tag{23}$$

Now we use proof by contradiction. We assume that there are $2m$ directions at $\mathbf{D}$ where the derivatives are zero, and we will prove that such points are of measure zero. Without loss of generality, we assume that the first $2m$ components $\frac{\partial G}{\partial x_1}, \frac{\partial G}{\partial x_2}, \ldots, \frac{\partial G}{\partial x_{2m}}$ are all zero at $\mathbf{D}$.

For the $n - 2m$ components with non-zero derivatives, let the first non-zero derivative be $\frac{\partial G}{\partial x_{2m+1}}$ with a derivative value of $d_1$. Since $G$ is a smooth function, we can choose a neighborhood $U_1$ of $\mathbf{D}$ such that for all $p \in U_1$, the absolute value of the derivative $\frac{\partial G}{\partial x_{2m+1}}$ at $p$ is greater than $\frac{d_1}{2}$. Similarly, for $x_{2m+2}$, let its derivative value be $d_2$, and we can choose $U_2$ such that for all $p \in U_2$, the absolute value of the derivative $\frac{\partial G}{\partial x_{2m+2}}$ is greater than $\frac{d_2}{2}$. Following this reasoning, we can obtain neighborhoods for all $n - 2m$ components, and we define $U = \bigcup_{i=1}^{n-2m} U_i$. We know that at any point in $U$, the derivatives of the last $n - 2m$ components remain non-zero.

Now consider the first $2m$ components. Since the derivatives of the last $n - 2m$ components are non-zero in the neighborhood, if we require that the points on the manifold with $2m$ components have zero derivatives, then the second derivatives of the first $2m$ components must also be zero. Specifically, for $x_1$, we need $\frac{\partial^2 G}{\partial x_j \partial x_1} = 0$ for $j = 1, 2, \ldots, 2m$. According to the expression in Eq. 23, if $\frac{\partial^2 G}{\partial x_j \partial x_1} = 0$, then $x_j$ can only take values at two points on either side of $x_1$. Similarly, the same conclusion applies to $x_2, x_3, \ldots, x_{2m}$. Therefore, it is easy to see that if the first $2m$ components can only take two values, the derivatives of these components will always be non-zero on the manifold, which is clearly a set of measure zero.

$\square$

Using the above lemma, we can prove that, given an $R$, the sample set in $D$ that can generate $R$ and contains samples identical to $R$ is a zero-measure set among all possible sample sets. The next step is to relax the condition of strict sample equality. In practical applications, we might consider that the RKME $Z$ includes samples that are very close to a particular sample in $D$. In this case, an attacker could use this sample as an approximation for the corresponding sample in $D$, thereby exposing the data in $D$ to risks from $Z$. Therefore, we want to define a tolerance level $\delta$ to determine whether two samples are sufficiently close. If the distance between two samples is less than $\delta$, we consider them close enough that privacy may be at risk. We aim to prove that, with high probability for a reasonable $\delta$, any point in RKME $Z$ is more than $\delta$ away from any point in $D$.

Thus, the discussion of the selection and rationale for $\delta$ is a key issue. It is important to note that $\delta$ must be a quantity related to the distribution, as the scale of the data will directly impact the choice of $\delta$. Larger-scale data can accommodate a larger $\delta$, while using the same $\delta$ standard for data with significantly different scales would be unreasonable. On the other hand, as the number of data points increases, the data becomes denser in feature space. In an extreme case, if we assume there are infinitely many data points, the data can be viewed as being present at every point in a continuous distribution. Thus, regardless of the form of the RKME $Z$, there will inevitably be data points that are identical to it. Therefore, our setting for $\delta$ should depend on the scale of the data and the amount of data. Next, we will discuss our settings for $\delta$ under three types of risks, along with the rationale behind these settings.

We will first discuss the choice of $\delta$ in the context of linkage and inference attacks, as these two types have more geometric intuitions. In a linkage attack, after obtaining the RKME $Z$, an attacker will use a brute-force attack to find all possible original sample sets $D$ that could generate $Z$. We need to consider that for a dataset $D$, if the minimum distance between two distinct data points is denoted as $d_{\min}$, then setting $\delta$ to $d_{\min}$ makes sense. This is because, in the original dataset, there are two data points at a distance of $d_{\min}$, and these two points are not the same. Therefore, we can consider that points in the original dataset that are $d_{\min}$ apart are indeed different. A natural choice for $\delta$ is $d_{\min}$. However, this choice is dependent on the dataset itself, and when two data points are very close to each other, this setting can easily weaken our conclusions. Starting from this perspective, we choose a stronger $\delta = \frac{L}{n}$, where $L$ is a measure of the dataset's scale, which we can take to be the range of the dataset. A natural observation is that $\frac{L}{n} > d_{\min}$. Therefore, if we can prove that this setting yields stronger conclusions than simply choosing $d_{\min}$, while also acknowledging the significance of $d_{\min}$, it follows that $\frac{L}{n}$ is also meaningful.

For inference attacks, we can similarly choose $\frac{M_{\text{infer}}}{n}$, where $M_{\text{infer}}$ is the range of the attribute we want to attack. It is important to note that our selection does not represent the maximum $\delta$ that satisfies our conclusions. Our primary focus is not on how far the synthetic data $Z$ generated by RKME can deviate from the original data, but rather on exploring the potential for RKME's false points, within a distance of $\delta$, to expose the privacy of the original data. We are concerned with the likelihood of privacy exposure under these conditions. In fact, we could derive a conclusion regarding $\delta$ as a variable, but this would affect the interpretability of our conclusions and is not what we require. The chosen $\delta$ has general applicability (as it is greater than $d_{\min}$), so we fix this $\delta$ to observe how the effectiveness of privacy protection relates to variations in data points. Another noteworthy point is that our conclusions regarding the validity of $\delta$ represent a worst-case scenario for all points in the sample. In reality, most points are likely to be much farther from RKME than $\delta$.

For consistency risk, since the definition of consistency risk is based on the distribution rather than specific datasets or RKME, a natural idea is to extend the aforementioned $\delta$ to its distributional version. We can achieve this extension using the range. Given a distribution with cumulative distribution function (CDF) $F(x)$, we know that the range can be defined as follows:

$$P\left(M_n \leq x\right) = F(x)^n$$
$$P\left(m_n \leq x\right) = 1 - (1 - F(x))^n$$

For a distribution where all moments exist, the expected value of the range can be approximated using the variance of the distribution. Thus, we select $\frac{\sigma}{n}$ as the value for $\delta$. In particular, if the second moment of the distribution does not exist, we can choose a range corresponding to a higher probability and then calculate the variance over the support of that region.

Before we present the lemma regarding the relaxation of $\delta$, we need a proposition:

**Proposition C.13.** *We have the following inequality:*

$$\int_{\mathcal{M}} \frac{d\mu}{\int_{\mathcal{M}} d\mu} \left( \mathbb{I}_{Z \cap D \neq \emptyset} \right) \leq \frac{1}{\binom{n}{2m}} C(n - 2m) \frac{\mathrm{Vol}(B_\delta)}{\mathrm{Vol}(B_L)}^{\frac{1}{n-2m}}.$$

*Proof.* We start from the idea of [Pan and Xu, 2009]. Let $\gamma$ be a $C^2$ closed and strictly convex plane curve with length $L$ and enclosing an area $A$, then

$$L^2 \leq 4\pi(A + |\tilde{A}|)$$

where $\tilde{A}$ denotes the oriented area of the locus of its curvature centers if and only if $\gamma$ is a circle.

Noticing that in the problem we are focusing on, we initially ignore the effect of $\delta$. In fact, what we aim to determine is the ratio of the perimeter of a lower-dimensional submanifold in its lower-dimensional space to the area of the manifold itself in the ambient space. This allows us to utilize ideas similar to the isoperimetric inequality mentioned earlier.

The first step is to perform local linearization at any given point. Based on our analysis in C.9 and C.12, we assume the curvature at this point is $C$. We denote the $2M$-dimensional linearization of the manifold $\mathcal{M}$ as $\mathcal{M}_{2M}$. Then, we have:

$$(\int_{\mathcal{M}} \frac{d\mu}{\int_{\mathcal{M}} d\mu} \left( \mathbb{I}_{Z \cap D \neq \emptyset} \right))_{2M} \bigcap B_D(\delta') \leq 2m(n - 2m) \frac{\mathrm{Vol}(B_\delta)}{\mathrm{Vol}(B_{\delta'})}^{\frac{1}{n-2m}}.$$

By continuity and bounded curvature, we first relax the local restriction, and we have:

$$(\int_{\mathcal{M}} \frac{d\mu}{\int_{\mathcal{M}} d\mu} \left( \mathbb{I}_{Z \cap D \neq \emptyset} \right))_{2M} \leq C2m(n - 2m) \frac{\mathrm{Vol}(B_\delta)}{\mathrm{Vol}(B_L)}^{\frac{1}{n-2m}}.$$

This represents the case of a $2m$-dimensional linear subspace in an $n$-dimensional space. Distinguishing all possible cases, we then have:

$$\int_{\mathcal{M}} \frac{d\mu}{\int_{\mathcal{M}} d\mu} \left( \mathbb{I}_{Z \cap D \neq \emptyset} \right) \leq \frac{1}{\binom{n}{2m}} C(n - 2m) \frac{\mathrm{Vol}(B_\delta)}{\mathrm{Vol}(B_L)}^{\frac{1}{n-2m}}.$$

$\square$

**Lemma C.14.** *Given an RKME $R$, let the set of all possible datasets $D$ that could generate this RKME correspond to points $\mathbf{D}$ in $\mathbb{R}^n$ denoted as $M$. We define the subset of $M$ where any component coordinate falls within $(z_i - \delta, z_i + \delta)$ for $i = 1, 2, \cdots, m$ as $M_Z$. Then the following inequality holds:*

$$\frac{\mathrm{Vol}(M_Z)}{\mathrm{Vol}(M)} \leq \frac{1}{\binom{n}{2m}} C_0(n - 2m) \frac{\mathrm{Vol}(B_\delta)}{\mathrm{Vol}(B_L)}^{\frac{1}{n-2m}}. \tag{24}$$

*Proof.* From Propositions C.9, C.12, and C.13, we obtain that given a connected component of a manifold, we have

$$\frac{\mathrm{Vol}(M_Z)}{\mathrm{Vol}(M)} \leq (n - 2m)C_0 \frac{\mathrm{Vol}(B_\delta)}{\mathrm{Vol}(B_L)}^{\frac{1}{n-2m}} \tag{25}$$

where $C_0$ is the ratio of $\mathrm{Vol}(\tilde{M})$ to $\mathrm{Vol}(M)$ in Lemma C.13. Since there may be multiple disconnected manifolds in the space, and the intervals $(z_i - \delta, z_i + \delta)$ may not intersect with some of these manifolds, the case where they do intersect corresponds to the number of $m$-dimensional subspaces in $n$-dimensional space. Therefore, the overall conclusion is

$$\frac{\mathrm{Vol}(M_Z)}{\mathrm{Vol}(M)} \leq \frac{1}{\binom{n}{2m}} C_0(n - 2m) \frac{\mathrm{Vol}(B_\delta)}{\mathrm{Vol}(B_L)}^{\frac{1}{n-2m}}. \tag{26}$$

$\square$

In the context of this paper, we regard brute-force attacks as the most potent form of assault on deterministic algorithms like RKME (Reduced Kernel Mean Embedding). Deterministic algorithms are characterized by producing consistent outputs for the same input. This principle is applicable even to common cryptographic methods such as elliptic curve encryption, where, under the assumption of sufficient computational resources, it's possible to decrypt the original message through exhaustive enumeration. However, the distinctive aspect of the deterministic algorithm discussed in this paper, particularly RKME, is its substantial data compression. This compression results in a scenario where a single output corresponds to numerous distinct inputs, diverging from typical deterministic models where one input maps to one output.

In Alg. 4.1 and Alg. 4.2, it is common practice to set the adversary's strategy as either a black-box or a white-box learning algorithm. Existing Membership Inference Attacks (MIAs) on generative models primarily concentrate on non-parametric deep learning models used for synthetic image generation. These studies largely explore the privacy risks associated with either model-specific white-box attacks or set membership attacks, which presuppose the adversary's access to the complete set of training records. The findings suggest that black-box MIAs, targeting specific records, perform only marginally better than random guessing. Regrettably, such prior attacks do not provide a reliable foundation for evaluating the privacy benefits of publishing synthetic data. Non-parametric models for non-tabular data represent a minimal range of use cases, and white-box attacks fail to accurately mirror the data sharing context. Moreover, set inference attacks are not apt for assessing privacy gains at an individual level.

For deterministic algorithms like the one we discuss, learning algorithms are unlikely to outperform brute-force attacks. The reason lies in the nature of the Reduced Kernel Mean Embedding (RKME) specification: for a single RKME specification, there could be infinitely many original datasets corresponding to it. However, among these infinite datasets, the ones containing a specific target record constitute only a set of measure zero. Calculating the area of such complex manifolds is already an NP-hard problem, making our focus on brute-force attacks universally applicable and a pragmatic choice for analysis. This approach acknowledges the inherent limitations of learning algorithms in predicting the exact dataset from a given RKME specification, due to the overwhelming diversity of potential original datasets.

### C.2 Proof of Proposition 3.2

*Proof.* From Lemmas C.2 and C.5, we can conclude that, except for a measure-zero set, all other sets $D$ in $\mathbb{R}^n$ satisfy the inequality in Eq. 4.3 with a non-zero lower bound. Therefore, there exists a unique $Z$ corresponding to it. $\qquad\square$

### C.3 Proof of Proposition 3.3

*Proof.* From Corollary C.11 and Lemma C.12, we immediately obtain that there exists a component that is the same as $Z$ and that $D$ is an $(n - 2m - 1)$-dimensional submanifold of the $n - 2m$-dimensional manifold. $\qquad\square$

### C.4 Proof of Theorem 3.4

*Proof.* From Propositions 3.2 and 3.3, we immediately obtain that the set of points $D$ in $\mathbb{R}^n$ where the generated RKME $Z$ satisfies $\exists i, j$ such that $z_i = y_j$ is Lebesgue null in $\mathbb{R}^n$. Furthermore, any continuous distribution is a continuous measure, and the null sets in composite spaces are also null sets in probability. $\qquad\square$

### C.5 Proof of Theorem 3.5 and Corollary 3.6

*Proof.* Through Lemma C.14, we immediately obtain three conclusions regarding risk: For consistency risk, we only need to take the expectation of Eq. 26 with respect to the distribution and select $\delta = \frac{\sigma}{n}$ and $\gamma = \frac{1}{2\sigma^2}$. We immediately get

$$R_C(\mathcal{P}) < \frac{C_0 \pi^{1/2}(n - 2m)}{\Gamma\left(\frac{n}{2} + 1\right) \cdot \Gamma\left(\frac{n-1}{2} + 1\right)^{-1}} \frac{1}{n}^{\frac{1}{n-2m}} = \mathcal{O}\left(\left(\frac{1}{e}\right)^{n-2m}\right) \qquad (27)$$

In particular, from the proof of Lemma C.10 and our choice of $\delta = \frac{\sigma}{n}$ and $\gamma = \frac{1}{2\sigma^2}$, we have $C_0 < 0.001$. At this step, Lemma C.9 requires $m \leq k\sqrt{n}$, where $k = d!$ is a constant. At this point, $\frac{\pi^{1/2}(n-2m)}{\Gamma\left(\frac{n}{2}+1\right)\cdot\Gamma\left(\frac{n-1}{2}+1\right)^{-1}}\frac{1}{n}^{\frac{1}{n-2m}} < 1$ (which holds for almost all $n > 5$). Therefore, $R_C < 0.001$. $\qquad\square$

## C.6 Proof of Theorem 4.2 and Corollary 4.5(Part 1)

*Proof.* For $R_L(Z)$, we can directly apply Lemma C.14 to obtain:

$$R_L(\mathcal{Z}) < C_0 \frac{dm!}{(dn-dm-2)!}\frac{1}{n}^{\frac{1}{dn-2dm}} = \mathcal{O}\left(\left(\frac{1}{e}\right)^{nd-2md-m}\right) = O\left(\frac{dm!}{(dn-dm)!}\right) \quad (28)$$

Likewise, by choosing $\delta = \frac{dL}{n}$, we have $R_L(Z) < 0.001$. The only requirement here is that when using Lemma C.10, we require $m \leq \frac{n}{2}$. $\qquad\square$

## C.7 Proof of Theorem 4.4 and Corollary 4.5(Part 2)

*Proof.* For $R_I(Z)$, the only difference from $R_L(Z)$ is that we are considering a one-dimensional manifold. Using Lemma C.14 we have

$$R_I(Z) < O\left(\frac{(2m)!}{e^{(n-2m-1)}(n-1)!}\right) \quad (29)$$

By choosing $\delta = \frac{L}{n}$, we have $R_I(Z) < 0.001$. The only requirement here is that when using Lemma C.10, we require $m \leq \frac{n}{2}$.

$\qquad\square$

## C.8 Proof of the Some Remarks

Now we prove several remarks mentioned in the text that need proof:

**Proposition C.15.** *The Laplacian kernel $k(x,y) = \exp(-\gamma\|x-y\|_1)$ cannot protect privacy like the Gaussian kernel. Specifically, it does not satisfy Theorem 3.4.*

*Proof.* The Laplacian kernel satisfies all the conclusions prior to Lemma C.12, and the analysis can be conducted similarly, with differences arising primarily from the expressions used for the various derivatives. However, the Laplacian kernel does not satisfy Lemma C.12, and its conclusions are in fact completely contrary to those of Lemma C.12. Assume that at point $p$ on the manifold $M$, the derivatives of the first $m$ components are zero. Notably, when $x_i$ is determined, the derivative $k(x,x_i)$ at $x_i$ is non-smooth for the Laplacian kernel, exhibiting jumps in different derivative directions. If perturbations are allowed, as long as the perturbation does not exceed this range in its effect on the derivatives, the first $m$ components will still equal zero in a small neighborhood around point $p$. This leads to the conclusion that the manifold still has dimensions of $n - 2m$ after fixing components and using the coordinate axis to slice through the manifold, which may not necessarily be a set of measure zero on $M$. $\qquad\square$

The significance of this proposition lies in demonstrating that the privacy protection capability of RKME is not solely a result of the reduction process, as not all kernels can ensure privacy protection after generating RKME through the reduce set approach. On one hand, this supports the rationale for our choice of the Gaussian kernel; on the other hand, it raises a future work question: what types of kernels can protect privacy? Or, how should we select kernels? Comparing the Laplacian and Gaussian kernels, both exhibit non-rationality, but they differ in terms of smoothness. Do all smooth kernels have the potential to protect privacy? This will be further discussed in our future work.

**Proposition C.16.** *For discrete distributions, we have the following conclusion: we represent a discrete distribution with a probability measure greater than zero at $k > m$ points using points in $\mathbb{R}^k$. Thus, $\mathbb{R}^k$ represents the family of all discrete distributions with a probability measure greater than zero at $k$ points. Almost all distributions in $\mathbb{R}^k$ satisfy Theorem 3.4.*

*Proof.* Based on Corollary C.11 and Lemma C.12, we immediately obtain this conclusion. $\qquad\square$

It's important to note that the only difference with discrete distributions is that while the theorem holds for any continuous distribution, it only applies to almost all distributions within the family for discrete distributions. This is due to two reasons: first, since the measure at any single point of a discrete distribution is always greater than zero, if the possible values of the discrete distribution include a point in $\mathbb{R}^n$ that could expose privacy, then the possibility of privacy exposure is non-zero. This characteristic prevents us from making the theorem universally applicable to all distributions. However, if we consider a family of distributions, we can still demonstrate a useful conclusion: for almost all distributions, Theorem 3.4 remains valid.

Secondly, when the number of points for a discrete distribution is less than $m$, any combination of the discrete distribution can certainly be represented by RKME, which would inevitably lead to privacy exposure. Specifically, if the discrete distribution is a binomial distribution with a size of 2 in RKME, then the RKME points will correspond to those two points of the discrete distribution, and the coefficients $\beta$ will reflect the counts of those two points in the dataset, since at this point the difference between RKME and the empirical KME of the dataset is zero. Therefore, this necessitates additional discussion regarding discrete distributions.

# D   A simple validation experiment

We have conducted validation experiments to further illustrate the tradeoff between data privacy and search quality in our work. Below, we present the experimental setting and empirical results. It's important to note that this paper is a theoretical discussion on the privacy protection capabilities of RKME specifications and does not propose any new algorithms. Instead, it validates the theories presented in the paper using existing algorithms. The method for constructing the learnware market comes from [Liu et al., 2024].

**Datasets.**   We use six real-world datasets: Postures [Gardner et al., 2014], Bank [Moro et al., 2014], Mushroom [Wagner et al., 2021], PPG-DaLiA [Reiss et al., 2019], PFS [Kaggle, 2018], and M5 [Makridakis et al., 2022]. These datasets cover six real-world scenarios involving classification and regression tasks. Postures involves hand postures, Bank relates to marketing campaigns of a banking institution, and Mushroom contains different mushrooms. PPG-DaLiA focuses on heart rate estimation, while PFS and M5 concern sales prediction. These datasets span various tasks and scenarios, varying in scale from 550 thousand to 46 million instances.

**Learnware market.**   We have developed a learnware market prototype comprising about 4000 models of various types. We naturally split each dataset into multiple parts with different data distributions based on categorical attributes, and each part is then further subdivided into training and test sets. For each training set, we train various models with different model types, including linear models, LightGBM, neural networks with different hyperparameters, and other common models. The number of models in each scenario ranges from 200 to 1500. For evaluation, we use each test set as user testing data, which does not appear in any model's training data. The various scenarios, partitions, and models ensure that the market encompasses a wide array of tasks and models, significantly enhancing the diversity in the prototype and the authenticity of experimental settings.

**Evaluation.**   We explored the tradeoff between data privacy and search ability in the six scenarios mentioned above. For search ability, a natural metric is to evaluate the performance of the model obtained through the search on the user's dataset. Good performance indicates that we have found a more suitable model. Therefore, we employ error rate and root-mean-square error (RMSE) as the loss function for classification and regression scenarios, respectively, collectively referred to as Search error. A smaller search error indicates stronger search ability.

For data privacy, we calculate the empirical risk for the three types of privacy risks proposed in this paper. Consistency risk is defined as $1 - \widehat{R}_C$, where $\widehat{R}_C$ is the sample estimate of $R_C$ in the paper, defined as the number of samples in the generated RKME synthetic data that are close to the original samples in terms of the Euclidean norm. Linkage and Inference risks are defined as $\widehat{R}_L(D) - \widehat{R}_L(Z)$ and $\widehat{R}_I(D) - \widehat{R}_I(Z)$, respectively, where $\widehat{R}_L(D), \widehat{R}_I(D), \widehat{R}_L(Z)$, and $\widehat{R}_I(Z)$

represent the confidence given by a brute force attack on the dataset $D$ or RKME $Z$. Smaller privacy risks indicates stronger data preservation ability.

**Configuration.** For the specification of RKME, we use a Gaussian kernel $k\left(\boldsymbol{x}_1, \boldsymbol{x}_2\right) = \exp\left(-\gamma\left|\boldsymbol{x}_1 - \boldsymbol{x}_2\right|_2^2\right)$ with $\gamma = 0.1$. For all user testing data, we set the number of synthetic data points in RKME, $m$, to $0, 10, 50, 100, 200, 500$, and $1000$ to explore the tradeoff between search ability and data privacy (when $m$ is $0$, a model is randomly selected). Our detailed experimental results can be found in the accompanying PDF. We summarize some representative results in the following table:

| | | Posture | Bank | MR | PPG | PFS | M5 |
|---|---|---|---|---|---|---|---|
| | $m = 10$ | 43.57% | 15.58% | 32.55% | 31.98 | 2.41 | 2.33 |
| Search error | $m = 100$ | 23.43% | 14.13% | 16.29% | 20.62 | 2.18 | 2.19 |
| | $m = 1000$ | 21.15% | 13.97% | 15.36% | 18.81 | 2.21 | 2.07 |
| | $m = 10$ | 0.000% | 0.000% | 0.000% | 0.000% | 0.000% | 0.000% |
| Consistency risk | $m = 100$ | 0.001% | 0.000% | 0.001% | 0.003% | 0.000% | 0.002% |
| | $m = 1000$ | 0.041% | 0.038% | 0.039% | 0.040% | 0.047% | 0.036% |
| | $m = 10$ | 0.01% | 0.02% | 0.02% | 0.03% | 0.02% | 0.02% |
| Linkage risk | $m = 100$ | 0.16% | 0.18% | 0.15% | 0.19% | 0.14% | 0.15% |
| | $m = 1000$ | 0.30% | 0.34% | 0.31% | 0.32% | 0.33% | 0.37% |
| | $m = 10$ | 0.01% | 0.01% | 0.01% | 0.02% | 0.01% | 0.02% |
| Inference risk | $m = 100$ | 0.11% | 0.10% | 0.18% | 0.13% | 0.12% | 0.14% |
| | $m = 1000$ | 0.42% | 0.40% | 0.46% | 0.41% | 0.39% | 0.43% |

It can be observed that as the number of synthetic data points $m$ in RKME increases, the search error decreases. This indicates that more synthetic data leads to better search ability. At the same time, as $m$ increases, all three privacy risks also increase, indicating that more synthetic data may lead to greater privacy risks. It is noted that as the number of synthetic data points $m$ in RKME increases, the search error initially decreases rapidly, but the rate of decrease slows down after $m = 100$. Conversely, the three privacy risks initially increase slowly but then rise more sharply after $m = 100$. Given that the number of user test data points $n$ we used ranges from 10,000 to 100,000, this aligns with our theoretical expectations in the paper that $m \in [\sqrt{n}, k\sqrt{n}]$.

