# OpenReview forum: "On the Ability of Developers' Training Data Preservation of Learnware"
_NeurIPS.cc/2024/Conference — NeurIPS 2024 poster_

### Official Review · Reviewer_4E2V · 2024-07-07

**Soundness:** 3
**Presentation:** 2
**Contribution:** 3
**Rating:** 7
**Confidence:** 2

**Summary:**

The authors theoretically analyze the properties of the learnware paradigm. In the learnware paradigm, a model developer can provide their trained models for other developers to use. To enable re-use, along with the model the developer provides a model specification that adequately represents the model's training data. This allows developers looking for models to find those that are most useful for their tasks of interest. Importantly, this specification should preserve the privacy of the original training data of the model.

While, the reduced kernel mean embedding specification has been proposed in the literature, a theoretical analysis that guarantees the protection of the model's training data is missing. The authors prove that RKME can simultaneously have the following three desirable properties:
* It does not contain any of the original training data points
* It is robust against inference attacks
* It still preserves sufficeint information about the training data for effective use as a learnware specification.

**Strengths:**

To the best of my knowledge, the results provided by the authors are novel. While I am not an expert neither in learnware systems nor in reproducing kernels, the results provided and the tools used in the analysis are non trivial. The result should be of high significance to the learnware community, especially since disclosing a model specification may carry risk if there are no formal guarantees. In terms of writing, the authors introduce the learnware problem and their contribution in a clear manner in Sections 1 and 2. The figures presenting the trade-offs between the different choices of $m$ are also very helpful for readers who may not be able to follow the theoretical results.

**Weaknesses:**

I think the clarity of Sections 3 and 4 can be significantly improved, so that they can be more approachable to a broader audience.

For Section 3, the authors present core results that are the basis of Theorems 3.4 and 3.5 but the connection to these theorems is not particularly clear. I would advise the authors to first explain the proof sketch and then present the key lemmas and how they connect to the proof sketch. See also the questions section for more.

For Section 4, I understand that the setting is even more complicated compared to Section 3 but providing some more intuition behind Theorem 4.2, especially the parts that are not already covered in Section 3, would also be helpful.

**Questions:**

I am a bit confused with regards to Theorem 3.4: Is Theorem 3.4 proven for the $\delta=0$ case or is it proven for a specific $\delta$? Intuitively, the overlap of a continuous distribution and a discrete distribution of synthetic data should be zero for $\delta=0$ regardless of how the number of discrete synthetic points. So I feel I am missing something.

Also I am still not sure which is the $\delta$ chosen for Theorem 3.5. Can you explain this more?

I am a bit confused about the linkability privacy game. It seems like the game can be technically split in two games, one when $b=0$ and one when $b=1$. Given that the adversary knows $b$, these two subgames are completely independent. In addition, the subgame of $b=0$ is trivial because the adversary trivially knows the answer. I am thus unsure what is the value of having the $b=0$ subgame at all. I guess my question here, is why is the random $b$ introduced in the game?

**Limitations:**

The authors have addressed the limitations.

---

> ### Author Rebuttal · Authors · 2024-08-07
>
> Many thanks for the constructive reviews!
>
> ---
>
> Q1: I am a bit confused with regards to Theorem 3.4: Is Theorem 3.4 proven for the $\delta=0$ case or is it proven for a specific $\delta$ ? Intuitively, the overlap of a continuous distribution and a discrete distribution of synthetic data should be zero for $\delta=0$ regardless of how the number of discrete synthetic points. So I feel I am missing something.
>
> A1: Thank you for your thorough review of Theorem 3.4 and for raising specific questions! Indeed, Theorem 3.4 is proven for the $\delta = 0$ case, but this is not a trivial case:
>
> In our paper, the consistency risk is defined as $R\_C(\mathcal{P}) \triangleq \mathbb{E}\_{D \sim \mathcal{P}^n}\left(\mathbb{I}\_{Z\_\delta \cap D\_\delta \neq \emptyset}\right)$. Therefore, we are concerned not with the overlap of a continuous distribution and a discrete distribution of synthetic data, but with the probability that a dataset sampled from the continuous distribution overlaps with the synthetic data generated from that dataset. The value of $R_C(\mathcal{P})$ at $\delta = 0$ strongly depends on the properties of method of generating synthetic data. For example, if we replace the commonly used Gaussian kernel in RKME, $k\left(\boldsymbol{x}_1, \boldsymbol{x}_2\right)=\exp \left(-\gamma\left|\boldsymbol{x}_1-\boldsymbol{x}_2\right|_2^2\right)$, with a Laplacian kernel, $k\left(\boldsymbol{x}_1, \boldsymbol{x}_2\right)=\exp \left(-\gamma\left|\boldsymbol{x}_1-\boldsymbol{x}_2\right|\right)$, it can be shown through analysis of its nonlinear equations that for any dataset, the generated synthetic data in RKME will always contain points from the original dataset. In this case, even if $\delta = 0$, the $R_C(\mathcal{P})$ for RKME with the Laplacian kernel would be $1$. Therefore, the proof in our paper that $R_C(\mathcal{P})$ for RKME with the Gaussian kernel is $0$ when $\delta = 0$ is non-trivial and relies on the analysis of nonlinear equations involving the Gaussian kernel.
>
> In the paper, we separately prove the $\delta = 0$ case because it has some geometric intuition and provides insights for the proof of the $\delta > 0$ case. In future versions, we will make the significance and position of this example clearer in the text.
>
> ---
>
> Q2: Also I am still not sure which is the δ chosen for Theorem 3.5. Can you explain this more?
>
> A2: Thanks for your valuable feedback!
>
> A detailed discussion on the selection of $\delta$ can be found in Appendix D. Here, we provide a simple understanding and discussion. The meaning of $\delta$ is that we want the synthetic data to differ from the real data by at least $\delta$ to ensure that the real data does not appear in the synthetic data. Therefore, $\delta$ is related to the scale of the data. When the overall data scale is small, it is impossible to ensure a $\delta$ larger than the data scale to ensure privacy. Additionally, $\delta$ is related to the number of real data points. For example, when there are many real data points, they become very dense in $\mathbb{R}^d$, and the distance between real data points becomes very small. In this case, the feasible $\delta$ will also become correspondingly smaller. Hence, in the proof of Theorem 3.5, $\delta$ is chosen as $\frac{R}{n}$, where $R$ is the upper bound of the data and $n$ is the number of samples.
>
> A simpler and more relaxed form of this selection is to directly choose the smallest different sample distance in the sample set, as it is easily shown to be less than $\frac{R}{n}$. However, this approach provides users with a clear metric: they can use the smallest sample distance in their data set to measure the strength of privacy protection we can offer. Therefore, we have provided this easier-to-understand form in the paper. In future versions, we will provide a more detailed explanation of the selection of $\delta$ in Section 3.
>
> ---
>
> Q3: I am a bit confused about the linkability privacy game. ... why is the random $b$ introduced in the game?
>
> A3: Thanks for your feedback!
>
> The cases $b=0$ and $b=1$ indeed represent two completely independent subgames, and the random bit $b$ is a common and necessary setup in the privacy game. The random bit $b$ serves to indicate to the adversary whether the dataset provided is the real dataset. Without the random bit $b$, the adversary would still attempt to attack using some method even if they obtained the original dataset. Many membership inference attacks (e.g., black-box attacks) use a confidence estimation approach, and even with the original dataset, they cannot always provide a completely correct conclusion. In the linkage privacy game, we naturally want the adversary to be able to accurately determine conclusions if they have the original dataset. Therefore, we include a random bit $b$ to be sent to the adversary along with the dataset.
>
> Moreover, the random bit $b$ plays a crucial role in the subsequent inference privacy game, as obtaining the original dataset does not necessarily lead to correct attribute inference (there could be samples with completely overlapping attributes except for the target attribute).
>
> ---
>
> Q4: I think the clarity of Sections 3 and 4 can be significantly improved, so that they can be more approachable to a broader audience.
>
> A4: Thanks for the constructive feedback!
>
> We will address the two points mentioned by the reviewer in future versions as follows:
>
> 1. We will revise the order of Section 3, presenting the proof sketch first to give readers a preliminary understanding, followed by a detailed proof according to its logic.
> 2. We will provide more intuition in Sections 3 and 4, and move some of the more complex parts to the appendix as appropriate.
>
> ---
>
> In the end, we take this opportunity to sincerely thank you for the careful review, your suggestions are very insightful and important for further improving the paper. We are happy to provide further clarifications if needed in the following author-reviewer discussions.

---

> > ### Comment · Reviewer_4E2V · 2024-08-12
> > **The authors have answered my questions**
> >
> > The authors have answered my questions and the justifications provided would help significantly improve the paper if they are indeed included. Also the extensive experimental evaluations are appreciated on top of the paper's theoretical contributions. I thus increase my score to a 7.

---

> > > ### Author Response · Authors · 2024-08-12
> > >
> > > Dear Reviewer 4E2V,
> > >
> > > Thank you very much for your kind reply! We will revise our paper according to the constructive reviews.
> > >
> > > Best
> > >
> > > Authors

---

### Official Review · Reviewer_gkER · 2024-07-11

**Soundness:** 3
**Presentation:** 3
**Contribution:** 3
**Rating:** 6
**Confidence:** 2

**Summary:**

The paper presents the "Reduced Kernel Mean Embedding (RKME)" specification, which represents a model's capabilities while ideally preserving the privacy of the original training data. The paper provides a theoretical analysis and proves that the RKME specification can protect the training data against common inference attacks and maintain its utility in the learnware search process.

**Strengths:**

* This paper aims to resolve the crucial data privacy challenge while enabling the effective reuse of pre-trained models under the learnware setting.
* This paper provides a comprehensive theoretical framework to prove the efficacy of RKME in preserving privacy. The proofs are detailed and robust, offering a strong theoretical foundation for the claims about data privacy and security against inference attacks.
* The paper also discusses the practical implementation of the RKME specification in learnware systems.

**Weaknesses:**

* The paper focuses on theoretical proofs and lacks extensive empirical evidence to support the effectiveness of the RKME specification in real-world scenarios.
* The analysis primarily hinges on the assumption that the RKME specification works optimally with certain types of data distributions and kernel functions.

**Questions:**

I do not have particular questions as I am unfamiliar with the field.

**Limitations:**

The authors have included discussions about potential limitations.

---

> ### Author Rebuttal · Authors · 2024-08-07
>
> Many thanks for the constructive reviews!
>
> ---
>
> Q1: The paper focuses on theoretical proofs and lacks extensive empirical evidence to support the effectiveness of the RKME specification in real-world scenarios.
>
> A1: Thanks for the feedback! We are not entirely sure whether the "effectiveness of the RKME specification in real-world scenarios" you mentioned refers to the effectiveness of RKME in Learnware search or its effectiveness in protecting privacy while performing search tasks well. Therefore, we will provide a detailed discussion on both aspects below:
>
> 1. **For Learnware search**, in current learnware research, almost all data types that use RKME with Gaussian kernel as a specification have achieved very good results. [Liu et al., 2023] demonstrated through experiments that for tabular data, using RKME with Gaussian kernel for Learnware search significantly outperforms searching in the learnware market without a specification. Similarly, [Wu et al., 2023] have shown that image data, after feature extraction, can use RKME with Gaussian kernel and achieve good experimental results. It is important to note that the core contribution of our work is providing theoretical guarantees that RKME can protect user data privacy, while the effectiveness of RKME in Learnware model search is not within the scope of discussion of this paper.
>
> 2. **For the effectiveness of RKME in protecting data privacy in real-world scenarios**, we carried out detailed experiments, the results of which can be found in the global response to all reviewers. Additionally, we have provided an analysis of our experimental results in the global response to all reviewers. It is worth mentioning that our experimental results match our theory well and demonstrate that RKME effectively protects data privacy while performing Learnware search tasks efficiently.
>
> If you still have any concerns, we are looking forward to addressing any further questions during the reviewer-author discussion period!
>
> ---
>
> Q2: The analysis primarily hinges on the assumption that the RKME specification works optimally with certain types of data distributions and kernel functions.
>
> A2: Thanks for the feedback! Our theoretical analysis does not make non-trivial assumptions about data distributions, and our analysis is also applicable to a broad class of kernel functions. We chose the Gaussian kernel for our proofs because it is widely recognized as the best kernel for the MMD (maximum mean discrepancy) distance. Here is a detailed explanation:
>
> 1. Note that in proving Theorem 3.4, we show that it holds for any continuous distribution. In the subsequent remark following Theorem 3.4, on line 165, we further discuss the applicability of Theorem 3.4 to discrete distributions. The conclusion of Theorem 3.5 is derived based on Theorem 3.4, so the assumptions are consistent. Therefore, our theory is not limited to specific types of data distributions.
>
> 2. In learnware research, one of the most representative distribution metric, MMD(maximum mean discrepancy), is used to calculate distribution distance. MMD distance with characteristic kernel is commonly used in various scenarios, such as applying MMD in GAN on image data, and the gaussian kernel is the most commonly used. Thus, following the tradition in most field using MMD as distribution metric, we mainly conduct the theoretical analysis on RKME with gaussian kernel.
>
>    However, our analysis method is highly applicable to Kernels with *nonrationality* and *analyticity*. These properties determine whether the synthetic data in RKME are locally homeomorphic to the manifold of the sample space. Therefore, kernels that satisfy these properties (such as the Sigmoid kernel $K(x, y)=\tanh \left(\gamma x^T y+r\right)$, Cauchy kernel $K(x, y)=\frac{1}{1+\gamma|x-y|^2}$, etc.) can undergo a similar analysis and reach the same conclusion as Proposition 3.2 and Proposition 3.3.
>
>    For these kernels, the bound appearing in Theorem 3.5 should be estimated differently for each kerne For the Gaussian kernel, we used linearization and isoperimetric inequalities in our proof to estimate sub-manifold within the manifold formed by the samples, which is also where the difficulty of this problem lies. For other kernels, similar conclusions can be reached after selecting appropriate methods for estimation.
>
>
>
> ---
>
> We greatly appreciate the your feedback! We hope our responses have clarified our work. If you still have any concerns or feel unfamiliar with certain aspects, we are looking forward to addressing any further questions during the reviewer-author discussion period!

---

### Official Review · Reviewer_mMLd · 2024-07-14

**Soundness:** 3
**Presentation:** 3
**Contribution:** 3
**Rating:** 6
**Confidence:** 2

**Summary:**

The paper analyzes the data preserving properties of Learnware, wan interesting idea involving a marketplace of pretrained ML models. In Learnware, new inference tasks are matched to ML models capable of solving that task without any raw data being shared. Rather, the method leverages RKME to construct a smaller, synthetic representation of the model's distribution over inputs and outputs. In this work, the paper explores whether Learnware is secure against data privacy attacks (linkage, attribute inference) when using the Gaussian kernel and various assumptions on the data. More compact representations are shown to be harder to attack. However, this reduces model search (retrieval) quality inducing a tradeoff.

**Strengths:**

+ Analysis of the ability of Learnware to resist privacy attacks against the dataset used to train the model makes the Learnware ecosystem more robust. Demonstrating the tradeoff between privacy and search (retrieval) quality is an intuitively clear result.

+ The theoretical results and analyses seem novel to me, as far as I can tell. A brief search didn't turn up anything relevant. (However, this is outside my area of expertise so I'm unable to assess validity.)

**Weaknesses:**

- The paper analyzes the privacy-preserving properties of Learnware. However, I remain unsure about the benefits of the core Learnware system. Reading through the recent references ("Learnware: Small models do big"), I'm left with many questions which are not really addressed in any of the papers. I don't see how Learnware is better than the existing model sharing infrastructure (model hub, data and model cards, benchmark results, open-source training and inference code). The existing ML model sharing infrastructure is widely used already and doesn't require the new user to even label any data first. Please see the questions below.

- The Learnware ecosystem seems like a very niche area. Without additional details of system usage, it becomes difficult to assess the impact of contributions in this paper.

- I'm not really equipped to comment on the quality of the theoretical analyses. That said, the paper could do a better job of describing how the analyses build on and fit into the larger body of work on related tasks.

- Experiments exploring the tradeoff between data linkage protection and search performance would have been nice to have. Without these, I'm again left wondering if the existing ML model sharing infrastructure (which does not have this issue) is indeed better.

**Questions:**

- Is the Learnware market currently operational in the wild? Please provide additional details into which parts of the Learnware ecosystem are actually in use at this time vs hypothesized. For example, scale of daily uploads & downloads?

- Please describe how the Learnware approach outperforms existing ML model-sharing infrastructure (model hubs, data cards, model cards, benchmarks, open source training and inference routines). For example, why is downloading an image segmentation model checkpoint off a model hub after reading through its data and model cards, benchmark and performance reviews, and trying it out in the online UI insufficient? How well does Learnware's "anchor learnwares" mechanism work in this situation compared to the approach above?

- Which of the theoretical analyses or results included in this paper are novel compared to prior geometric analyses or privacy works? Please include references, if any, for the analytical techniques in the paper.

- Is it possible to experimentally explore the data linkage vs search quality tradeoffs? How does the search quality degradation affect the user experience of trying to find an appropriate model?

**Limitations:**

Yes

---

> ### Author Rebuttal · Authors · 2024-08-07
>
> Many thanks for the constructive reviews! We provide detailed responses below, and hope the reviewer could reassess the significance of our results. We are looking forward to addressing any further question in the reviewer-author discussion period.
>
> ---
>
> Q1: Is the Learnware market currently operational in the wild? ... For example, scale of daily uploads & downloads?
>
> A1: Thanks for the feedback!
>
> Yes. Recently, the first learnware dock system, *Beimingwu*, has been built and released [1,2]. With a novel specification-based architecture, the system can significantly streamline the process of building machine learning models for new tasks, by automatically identifying useful models. The system provides implementations and extensibility for the entire process of the learnware paradigm, including the submission, usability testing, organization, identification, deployment, and reuse of learnwares. The core engine is also released as *learnware* package [3].
>
> The Beimingwu system is currently primarily serving as a research platform for learnware studies. Although it is only open to the academic community, it has already been registered by over 500 researchers from more than **150 universities**.
>
> [1] Beimingwu: A Learnware Dock System. KDD 2024.
>
> [2] Website: https://bmwu.cloud/
>
> [3] Learnware package: https://learnware.readthedocs.io/
>
> ---
>
> Q2: Please describe how the Learnware approach outperforms existing ML model-sharing infrastructure ...   reading through its data and model cards, benchmark and performance reviews, and trying it out in the online UI insufficient?
>
> A2: Thanks for the detailed feedback!
>
> The biggest advantage of Learnware compared to existing ML model-sharing infrastructures lies in **model search with specification**. As you mentioned in your example, reading through data and model cards, benchmarks, and performance reviews can certainly help determine a good model, but when there are lots of models in the library, it is difficult to try each one individually, which necessitates model search. General existing ML model-sharing infrastructures can only perform keyword or language description searches. Since they cannot access specific user task data, such searches are imprecise and require users to try the search results themselves.
>
> In Learnware, besides using language description information (which we refer to as **semantical specification**), we also use the RKME-based **statistical specification** mentioned in this paper to describe the user's specific task for precise model localization. This allows users to search for models in the market that are more suitable for their specific tasks.
>
> To our best knowledge, for the first time, the learnware paradigm formally proposes to build a large model platform consisting of numerous high-performing models with specifications, and enable users to easily leverage existing models to solve their tasks. Recently, utilizing the large model platform to solve new learning tasks has witnessed a rapidly increasing attention, notably the Hugging Face platform, hosting over half a million models. Thus identifying truly helpful models becomes more and more difficult. Based on a novel statistical-specification-based architecture, learnware system aims to automatically identify and assemble high-performing models suitable for user tasks, with no need for extensive data and expert knowledge, while preserving data privacy we proved in this paper.
>
> ---
>
> Q3: The paper could do a better job of describing how the analyses build on and fit into the larger body of work on related tasks. Which of the theoretical analyses are novel compared to prior geometric analyses or privacy works?
>
> A3: Thanks for the feedback. To the best of our knowledge, our work is the first to use geometric analyses to study privacy. The theories and related analytical methods are completely original and provide significant contributions to the field of privacy:
>
> 1. The problem we mainly focus on is the privacy of synthetic data brought by data compression. However, related work (e.g., data distillation/data condensation) mainly explores privacy properties through experiments without theoretical guarantees. This is because the most mainstream privacy theoretical analysis framework, differential privacy (DP), is difficult to apply to compressed synthetic datasets. Thus, theoretical analysis methods for the privacy of compressed data have always been lacking. Our paper provides the first theoretical analysis attempt for RKME, a special form of data compression.
> 2. The essential difficulty in analyzing the privacy of RKME-compressed synthetic datasets lies in the fact that RKME is the optimal solution to a non-convex nonlinear system of equations involving a Gaussian kernel. Solving the optimal solution for non-convex problems involving Gaussian kernels has been an open problem. However, in geometric analyses, we found a way to analyze the properties of the solution without solving this non-convex nonlinear system. Our newly proposed technique may prove to be highly useful in analyzing the properties of solutions to specific forms of non-convex problems.
>
> ---
>
> Q4: Is it possible to experimentally ... to find an appropriate model?
>
> A4: Thanks for your constructive feedback! We have carried out detailed experiments, the datasets, settings, and results of which can be found in the global response to all reviewers. We have also analyzed the experimental results in the global response to all reviewers, and the empirical findings align closely with our theories.
>
> ---
>
> We once again thank you for your constructive comments! We believe that the Learnware paradigm is extremely important and general, and our research occupies a non-negligible position in both the learnware and privacy communities. We hope the above replies will address your concerns and would appreciate a reevaluation of our paper's score!

---

> > ### Comment · Reviewer_mMLd · 2024-08-13
> > **Re. author response**
> >
> > I thank the authors for their detailed response to the reviewers. A number of my concerns were about the underlying assumptions and properties of the Learnware system and these have been satisfactorily addressed in the response. At this point, I have no major objections to this paper and have increased my score to reflect this.

---

> > > ### Author Response · Authors · 2024-08-13
> > >
> > > Dear Reviewer mMLd,
> > >
> > > Thank you so much for your kind reply and for adjusting the score! We will revise our paper according to the constructive reviews.
> > >
> > > Best
> > >
> > > Authors

---

### Author Rebuttal · Authors · 2024-08-07

We have conducted validation experiments to further illustrate the tradeoff between data privacy and search quality in our work. Below, we present the experimental setting and empirical results. All related figures can be found in the accompanying PDF.

---

### **Datasets**

We use six real-world datasets: Postures [Gardner et al. 2014], Bank [Moro, Cortez, and Rita 2014], Mushroom [Wagner, Heider, and Hattab 2021], PPG-DaLiA [Reiss et al. 2019], PFS [Kaggle 2018], and M5 [Makridakis, Spiliotis, and Assimakopoulos 2022]. These datasets cover six real-world scenarios involving classification and regression tasks. Postures involves hand postures, Bank relates to marketing campaigns of a banking institution, and Mushroom contains different mushrooms. PPG-DaLiA focuses on heart rate estimation, while PFS and M5 concern sales prediction. These datasets span various tasks and scenarios, varying in scale from **550 thousand to 46 million** instances.

---

### **Learnware market**

We have developed a learnware market prototype comprising about **4000 models** of various types. We naturally split each dataset into multiple parts with different data distributions based on categorical attributes, and each part is then further subdivided into training and test sets. For each training set, we train various models with different model types, including linear models, LightGBM, neural networks with different hyperparameters, and other common models. The number of models in each scenario ranges from 200 to 1500. For evaluation, we use each test set as user testing data, which does not appear in any model’s training data. The various scenarios, partitions, and models ensure that the market encompasses a wide array of tasks and models, significantly enhancing the diversity in the prototype and the authenticity of experimental settings.

---

### **Evaluation**

We explored the tradeoff between data privacy and search ability in the six scenarios mentioned above. For search ability, a natural metric is to evaluate the performance of the model obtained through the search on the user’s dataset. Good performance indicates that we have found a more suitable model. Therefore, we employ error rate and root-mean-square error (RMSE) as the loss function for classification and regression scenarios, respectively, collectively referred to as Search error. A **smaller search error** indicates **stronger search ability**.

For data privacy, we calculate the empirical risk for the three types of privacy risks proposed in this paper. Consistency risk is defined as $1-\widehat{R}_C$, where $\widehat{R}_C$ is the sample estimate of $R_C$ in the paper, defined as the number of samples in the generated RKME synthetic data that are close to the original samples in terms of the Euclidean norm. Linkage and Inference risks are defined as $\widehat{R}_L(D) - \widehat{R}_L(Z)$ and $\widehat{R}_I(D) - \widehat{R}_I(Z)$, respectively, where $\widehat{R}_L(D)$, $\widehat{R}_I(D)$, $\widehat{R}_L(Z)$, and $\widehat{R}_I(Z)$ represent the confidence given by a brute force attack on the dataset $D$ or RKME $Z$. **Smaller privacy risks** indicates **stronger data preservation ability**.

---

### **Configuration**

For the specification of RKME, we use a Gaussian kernel $k\left(\boldsymbol{x}_1, \boldsymbol{x}_2\right)=\exp \left(-\gamma\left|\boldsymbol{x}_1-\boldsymbol{x}_2\right|_2^2\right)$ with $\gamma=0.1$. For all user testing data, we set the number of synthetic data points in RKME, $m$, to $0$, $10$, $50$, $100$, $200$, $500$, and $1000$ to explore the tradeoff between search ability and data privacy (when $m$ is 0, a model is randomly selected).

Our detailed experimental results can be found in the accompanying PDF. We summarize some representative results in the following table:



|                                 |          | Posture | **Bank** | **MR** | **PPG** | **PFS** | **M5** |
| ------------------------------- | -------- | -------- | -------- | ------ | ------------- | ------- | ------ |
| Search error  | $m=10$   | 43.57%| 15.58%| 32.55% | 31.98| 2.41| 2.33|
|              | $m=100$  | 23.43%| 14.13%|16.29%| 20.62| 2.18| 2.19|
|                 | $m=1000$ | 21.15%   | 13.97%   | 15.36% | 18.81| 2.21| 2.07|
| Consistency risk      | $m=10$| 0.000‰| 0.000‰| 0.000‰ | 0.000‰| 0.000‰  | 0.000‰ |
|                    | $m=100$  | 0.001‰| 0.000‰| 0.001‰ | 0.003‰| 0.000‰  | 0.002‰ |
|                                 | $m=1000$ | 0.041‰| 0.038‰| 0.039‰ | 0.040‰| 0.047‰  | 0.036‰ |
| Linkage risk| $m=10$   | 0.01‰| 0.02‰| 0.02‰  | 0.03‰| 0.02‰   | 0.02‰  |
|                                 | $m=100$  | 0.16‰| 0.18‰| 0.15‰  | 0.19‰| 0.14‰   | 0.15‰  |
|                                 | $m=1000$ | 0.30‰| 0.34‰| 0.31‰  | 0.32‰| 0.33‰   | 0.37‰  |
| Inference risk| $m=10$   | 0.01‰| 0.01‰    | 0.01‰  | 0.02‰| 0.01‰   | 0.02‰  |
|                                 | $m=100$  | 0.11‰| 0.10‰| 0.18‰  | 0.13‰| 0.12‰   | 0.14‰  |
|                                 | $m=1000$ | 0.42‰| 0.40‰| 0.46‰  | 0.41‰| 0.39‰   | 0.43‰  |



Due to the limitation of space, we provide a brief analysis of the experimental results as follows:

1. It can be observed that as the number of synthetic data points $m$ in RKME increases, the search error decreases. This indicates that more synthetic data leads to better search ability. At the same time, as $m$ increases, all three privacy risks also increase, indicating that more synthetic data may lead to greater privacy risks.
2. It is noted that as the number of synthetic data points $m$ in RKME increases, the search error initially decreases rapidly, but the rate of decrease slows down after $m = 100$. Conversely, the three privacy risks initially increase slowly but then rise more sharply after $m = 100$. Given that the number of user test data points $n$ we used ranges from 10,000 to 100,000, this aligns with our theoretical expectations in the paper that $m \in [\sqrt{n}, k\sqrt{n}]$.

---

### Decision · Program_Chairs · 2024-09-25

**Decision:**

Accept (poster)

**Comment:**

The paper puts forward a theoretical analysis of tradeoffs between quality and privacy in the use of pretrained models from a model hub (or "learnware marketplace") for new tasks, and in particular shows a promising theoretical result that a method using RKME to select component models is highly unllikely to leak data that those models were originally trained on.  The reviewers have reached a clear consensus that the work is technically sound.  There was significant discussion on the overall importance and impact of the work, and during the rebuttal phase the argument that this setting is important in the age of Hugging Face was found to be compelling -- and indeed I agree with this viewpoint.

Reviewers noted that empirical results would have strengthened the submission, and in the rebuttal phase the authors provided detailed and compelling empirical results on 6 datasets.  In general, I think it is difficult to weigh such results in the context of a review because they are a significant post-submission addition.  However, because they demonstrate results that were already proved theoretically (rather than supporting ideas that relied primarily on empirical validation for significance), I think this is acceptable.  These results should certainly be included in the final version, along with revisions and clarifications based on the reviews and the suggestions from reviewers.